# ComboStoc: Combinatorial Stochasticity for Diffusion Generative Models

## Abstract

In this paper, we study an under-explored but important factor of diffusion generative models, *i.e.*, the combinatorial complexity. Data samples are generally high-dimensional, and for various structured generation tasks, additional attributes are combined to associate with data samples. We show that the space spanned by the combination of dimensions and attributes is insufficiently sampled by existing training scheme of diffusion generative models, causing degraded test time performance. We present a simple fix to this problem by constructing stochastic processes that fully exploit the combinatorial structures, hence the name *ComboStoc*. Using this simple strategy, we show that network training is significantly accelerated across diverse data modalities, including images and 3D structured shapes. Moreover, *ComboStoc* enables a new way of test time generation which uses asynchronous time steps for different dimensions and attributes, thus allowing for varying degrees of control over them.

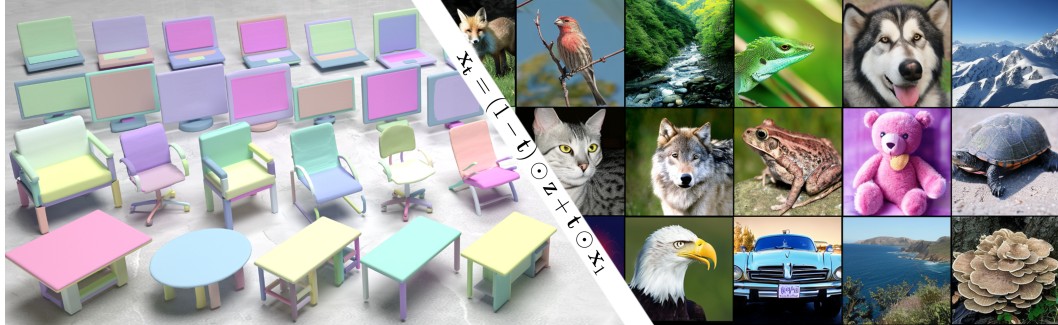

Figure 1: *ComboStoc* improves diffusion generative models across data modalities of images and structured 3D shapes. Left: structured 3D shapes where semantic parts are colored randomly. Right: images with consistently lower Frechet Inception Distance (FID) than baseline results. Middle: core to *ComboStoc* is a simple conversion of the interpolation schedule $t$ of the diffusion models into a tensor of the same shape as the data point $\mathbf{x}_1$ and noise point $\mathbf{z}$, and applying different values within [0, 1] for different dimensions or attributes to fully sample the combinatorial complexity of the dimensions and attributes.

## 1 Introduction

Diffusion generative models rely heavily on modeling the desired behavior over the whole space of possibilities, so that the generative models cover all data distributions systematically. However, the current training schemes generally focus on a single transport path from the source pure noise distribution to the target data distribution (Albergo & Vanden-Eijnden, 2023; Albergo et al., 2023; Liu et al., 2023; Lipman et al., 2023). The training therefore gives insufficient sampling of large regions of the entire space of possibilities, which nevertheless can be encountered because of stochastic sampling during evaluation and produce inaccurate behavior, leading to poor generation results.

To solve this mismatch between training scheme and test time evaluation, we propose to fully sample the space of combinatorial complexity. To see why the space of possibilities has a combinatorial

structure, we note that the data samples are most likely residing on high dimensional spaces with clear combinatorial structures. For example, the most powerful generative models so far use transformers as the network model (Peebles & Xie, 2023; Ma et al., 2024), and treat an image sample as a collection of patch tokens to be generated in parallel; moreover, each of the patch token is encoded as a vector of high dimensions. The combination of patches and their feature vectors present highly complex spaces, over which the diffusion generative models must learn to evolve toward data samples where patches and feature vectors are correlated nontrivially. In addition, for generative tasks in more structured domains, for example, 3D shapes with semantic parts, the combinatorial complexity is even more pronounced: each part has numerous attributes encoding different properties like its existence, bounding box and part shape, in addition to the part/patch decomposition and multiple feature channels analogous to images.

We sample the spaces of such combinatorial complexity by a simple modification of typical transport plans. In particular, instead of using a synchronized time schedule for each data sample, we apply asynchronous time steps for each of the patches/parts, attributes and feature vector dimensions, which allows for full sampling of a subspace spanning the various combinations of each pair of source and target data points.

| Model | Params(M) | Training Steps | FID |
|---|---|---|---|
| DiT-XL | 675 | 400K | 19.5 |
| SiT-XL | 675 | 400K | 17.2 |
| ComboStoc | 673 | 400K | **15.69** |
| DiT-XL | 675 | 800K | 14.3 |
| SiT-XL | 675 | 800K | 12.6 |
| ComboStoc | 673 | 800K | **11.41** |
| DiT-XL (cfg=1.5) | 675 | 7M | 2.27 |
| SiT-XL (cfg=1.5) | 675 | 7M | 2.06 |
| ComboStoc (cfg=1.5) | 673 | **800K** | 2.85 |

Table 1: Improvements over SiT across iterations.

We show that by simply enhancing the training scheme to incorporate the combinatorial sampling, the generative models for images and 3D structured shapes can be significantly improved (Fig. 1). In particular, for images from ImageNet, we obtain systematic FID-50k improvements along different training iterations than the baseline SiT model (Tab. 1). For 3D structured shapes which have even stronger combinatorial complexity, we show that our training scheme is indispensable for obtaining a working generative model.

In addition to the improved performances, the training scheme exploiting combinatorial stochasticity enables new modes of using the trained generative models. Specifically, we can now generate different patches/parts/attributes in asynchronous time schedules. This means that for example we can condition the final sample on flexible partial observations of a reference sample beyond binary masks. Instead, for images we can apply graded control across patches and channels. For structured shapes we can also specify the shapes of some parts only, and let the model generate the rest parts and attributes. These new modes of generation have the potential to unify specialized image and shape editing solutions.

## 2  BACKGROUND ON DIFFUSION GENERATIVE MODELS

The problem of generative modeling aims at capturing the complete distribution of a set of data samples. Its state-of-the-art solutions include denoising diffusion probabilistic models (Ho et al., 2020), score-based models (Song et al., 2021) and flow matching (Lipman et al., 2023; Liu et al., 2023), all of which transform a simple source distribution (*e.g.* the unit normal distribution) into the target distribution following the dynamics specified by variations of stochastic differential equations. Remarkably, the different formulations can be unified through the framework of stochastic interpolants (Albergo & Vanden-Eijnden, 2023; Albergo et al., 2023). In particular, the stochastic interpolants framework defines the process of turning data samples into source distributions and vice versa as a simple interpolation between the two distributions, augmented with random perturbations during the processes. Without loss of generality, we reproduce the formulation of a simple linear one-sided interpolant process below (illustrated in Fig. 2(a)):

$$\mathbf{x}_t = (1-t)\mathbf{z} + t\mathbf{x}_1, \; t \in [0, 1] \tag{1}$$

where $\mathbf{z} \sim N(0, \mathbf{1})$ samples the source distribution, $\mathbf{x}_1 \sim D$ samples the target data distribution, $t \in [0, 1]$ is the interpolation schedule. A network model $f_\theta(\mathbf{x}_t)$ can be trained to recover the interpolation velocity $\frac{\partial \mathbf{x}_t}{\partial t} = \mathbf{x}_1 - \mathbf{z}$, the target data sample $\mathbf{x}_1$, or the noise $\mathbf{z}$ (Albergo et al., 2023). To generate data samples, one then starts from a random sample $\mathbf{z}$, follows the velocity field and integrates them numerically into the final samples.

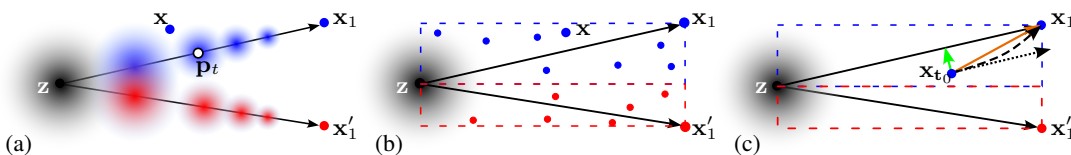

Figure 2: ***ComboStoc* enables better coverage of the whole path space.** Assuming two-dimensional data samples. **(a)** the standard linear one-sided interpolant model reduces its density as it approaches individual data samples; the low density regions are not well trained and once sampled would produce low-quality predictions. **(b)** using *ComboStoc*, for each pair of source and target sample points, a whole linear subspace spanned with their connection as the diagonal will be sufficiently sampled, so that there are fewer low-density regions not well trained. **(c)** when the network is trained to predict velocity $\mathbf{x}_1 - \mathbf{z}$ on an off-diagonal sample point $\mathbf{x_t}$, a compensation drift ($\mathbf{v}_{cmpn}$ in green) can be applied to pull the trajectory back to diagonal.

Remarkably, on modeling large scale image datasets like ImageNet, a scalable transformer architecture implementing the above process (Ma et al., 2024) shows state-of-the-art performance and outperforms alternative formulations, including DDPM (Ho et al., 2020) implemented via the same network (Peebles & Xie, 2023).

## 3   COMBINATORIAL STOCHASTIC PROCESS

Most interesting data samples are high dimensional. For example, state-of-the-art generative models encode images as latent patches with both spatial and feature dimensions (Ma et al., 2024; Peebles & Xie, 2023). 3D shapes structured as part ensembles include even more attributes in addition to spatial and feature dimensions, for example, the varying numbers of parts and their bounding boxes and positions (Mo et al., 2019b); to generate such data requires the handling of more flexible dimensions.

No matter how many dimensions and attributes a data sample has, standard diffusion generative models treat them homogeneously and in synchronization. For example, for the stochastic interpolants, the generative model is trained on samples distributed on densities with shrinking coverage along the transport paths connecting the source distribution and each target data sample, as illustrated in Fig. 2(a). In Appendix A.1, we provide a formal proof and visualization of the shrinking coverage. This design leaves the low density regions insufficiently trained, and once they are sampled in test stage via solving stochastic differential equations, the network tends to produce poor results.

To address the above problem, we emphasize the combinatorial complexity of individual dimensions and attributes of data samples. In particular, we purposefully sample points with asynchronous diffusion schedules of dimensions and attributes, as shown in Fig. 2(b). To implement the asynchronous schedules is rather simple. We turn the interpolation schedule $t$ of Eq. (1) into a tensor $\mathbf{t}$ of the same shape as $\mathbf{x}$, and use different values independently and uniformly sampled within $[0, 1]$ for the dimensions and attributes, to obtain the sample points:

$$\mathbf{x_t} = (1 - \mathbf{t}) \odot \mathbf{z} + \mathbf{t} \odot \mathbf{x}_1 \tag{2}$$

where $\odot$ is elementwise product. By construction, the sampling density is uniform within the subregions spanned by each pair of source and target data points. Correspondingly, the network $f_\theta(\mathbf{x_t})$ is trained to predict velocity, or the target data sample, etc.

The benefits of using these augmented samples from combinatorial stochasticity lie in three folds:

- We make sure the network coverage is broader than the synchronized schedule, so that during test stage the network performs more robustly and with higher quality.
- We encourage the network to learn the correlations of different dimensions and attributes, as the network is trained to synchronize them to reach the final data points.
- The trained network enables more flexible control over the generation process, where different dimensions and attributes can be given varying degrees of finalization to be synthesized in the final result.

Next, we discuss the detailed adaptations and achieved effects through generative tasks from two different domains, *i.e.* images and structured 3D shapes.

| (a) Image domain configurations | | |
|---|---|---|
| Feature ╱ Spatial | w/o patch | w/ patch |
| single code | unsync_none | unsync_patch |
| feature vector | unsync_vec | unsync_all |

| (b) Structured 3D shape configurations | | |
|---|---|---|
| Feature ╱ Spatial | w/o part | w/ part |
| single code | unsync_none | unsync_part |
| attribute | unsync_att | unsync_att_part |
| feature vector | unsync_vec | unsync_all |

Table 2: **Enumerating configurations of different combinatorial complexities**, for image domain generation (a) and structured 3D shape generation (b).

**Images.** For image generation, we take on the baseline of SiT (Ma et al., 2024) which applies highly scalable transformer networks and achieves state-of-the-art performance on ImageNet scale generation. In particular, a given image is encoded via the VAE encoder from Rombach et al. (2022) as a latent image $\mathbf{x}_1$ of shape $C \times H \times W$, and the network is trained to predict velocity given the diffused latent image $\mathbf{x}_t$ (Eq. (1)) and optionally conditioned on the image class $c$ and interpolation schedule $t$, *i.e.*, $f_\theta(\mathbf{x}_t; c, t) = \mathbf{x}_1 - \mathbf{z}$.

Correspondingly, we make several simple adaptations to implement the *ComboStoc* scheme. First, we construct $\mathbf{t}$ with the same shape of $C \times H \times W$, and update the timestep embedding module of SiT to accommodate this change (see Appendix A.3). Note that the conditioning on class labels and timesteps are mixed and implemented as modulation operations in Ma et al. (2024), and therefore are not symmetric to the data samples in importance[1]. Second, importantly, we note that for velocity prediction, the samples with asynchronous $\mathbf{t}$ should not predict the original velocity $\mathbf{x}_1 - \mathbf{z}$ only; otherwise there will be drift off the target data points during test stage integration (illustrated as Fig. 2(c) dotted line; see Appendix A.2 for a formal analysis). To mitigate this issue, we propose two possible approaches to compensate the velocity, as detailed in Appendix A.2. Due to limited computational resources, in our experiments, we have applied only the first approach of minimizing off-diagonal drift by gradient descent, and leave the test of the second approach for future work.

**Structured 3D shapes.** We use the generative modeling of structured 3D shapes as a new task to further demonstrate the importance of exploiting combinatorial complexity. Indeed, structure 3D shapes have even stronger combinatorial complexity than images, as shown in its varying numbers of parts, their positions and bounding boxes, as well as the detailed shape variations for each part. Precisely, we denote a structured 3D object as a collection of object parts, *i.e.*, $\mathbf{x} = \{\mathbf{p}_i\}, i \in [L]$, where we set $L = 256$ to cover the maximum number of parts in a dataset. An object part is further encoded as $\mathbf{p} = (s, \mathbf{b}, \mathbf{e})$, where $s \in [0, 1]$ indicates the existence of this part, $\mathbf{b} = (x, y, z, l, w, h)$ denotes the bounding box center $(x, y, z)$ and length $l$, width $w$ and height $h$, and $\mathbf{e} \in \mathbb{R}^{512}$ is a latent shape code encoding the part shape in normalized coordinates. Note that under this representation, a permutation of the part indices does not change the 3D shape, which is quite different from images represented as a feature grid of fixed order and size.

To generate structured 3D shapes with semantic parts, we train a stochastic interpolant model. In particular, given a structured 3D shape $\mathbf{x}_1 = \{\mathbf{p}_i\}$ and its diffused sample $\mathbf{x}_t$ (Eq. (2)), we make the network predict the target data sample directly for simplicity, *i.e.*, $f_\theta(\mathbf{x}_t; c, \mathbf{t}) = \mathbf{x}_1$, where $c$ is the optional class label of the 3D shape. Note that here $\mathbf{t}$ assigns different time steps for all the different attributes and dimensions of each object part. We validate the generative model for structured 3D shapes by training on the PartNet dataset (Mo et al., 2019b), as discussed in Sec. 4.

## 4 RESULTS AND DISCUSSION

In this part, we show that *ComboStoc* improves the training convergence of diffusion generative models for both images and structured 3D shapes. We also demonstrate the novel applications enabled by the asynchronous time steps of *ComboStoc*.

---

[1]The modulation by conditions including class labels and timesteps differ between training and test stages, as during training asynchronous timesteps are used while during testing synchronized timesteps will be used if no graded control is applied (Sec. 4.2). However, the training stage timesteps cover those of test stage as special cases, and thus enhance network generalization.

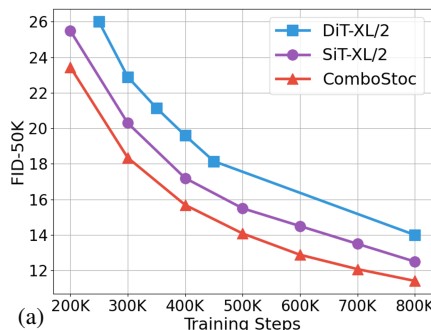 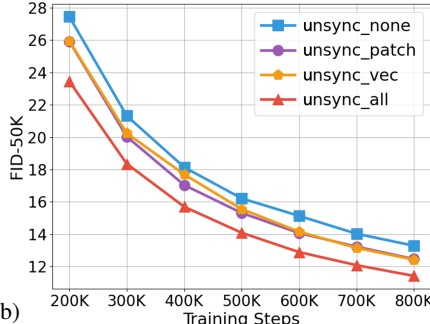

Figure 3: **Comparison on image generation with respect to training steps.** (a) plots the baseline SiT and our model, as well as DiT for reference; all models are of the scale XL/2 (Ma et al., 2024). (b) plots the different settings using varying degrees of combinatorial stochasticity.

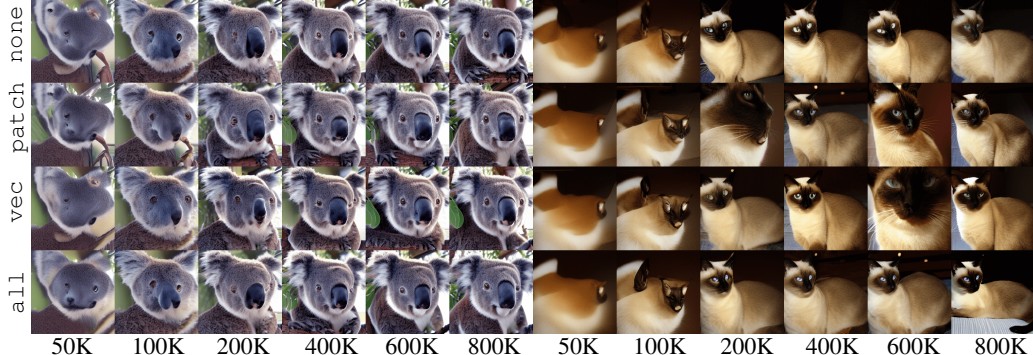

Figure 4: **Results of image generation at different training steps**. Settings with stronger combinatorial sampling produce well-structured images earlier; *e.g.* see the koala bear faces and cat eyes.

## 4.1 IMPROVED TRAINING OF DIFFUSION GENERATIVE MODELS

We explore the combinatorial complexities for both images and structured 3D shapes, and build corresponding configurations which exploit these complexities to compare with baseline configurations that do not apply asynchronous time schedules. We show that the different configurations improve over baseline configurations universally. In addition, we show that the higher degree of combinatorial complexity, the more important our scheme is for training a working model.

**Images**   Following SiT (Ma et al., 2024), we train on ImageNet (Deng et al., 2009) for class-conditioned image generation. To fully explore the effects of combinatorial stochasticity, we enumerate four settings with different levels of combinatorial flexibility in diffusing the data samples (see also Tab. 2(a)). In particular, we use `unsync_none`, `unsync_patch`, `unsync_vec`, and `unsync_all` to denote no splitting of time steps, using different time steps for latent image pixels, for latent image channels, and for both image pixels and channels. We run the different settings on top of the SiT-XL/2 baseline model. Considering the difficulty posed by ImageNet data size, in each batch we apply the split time steps only to half of the samples and leave the other half unchanged with synchronized time steps, which balances between samples along and off diagonal paths (Fig. 2)[2]. Plots of FID-50K (Heusel et al., 2017) with respect to training steps are shown in Fig. 3, where classifier-free guidance is not used.

First, as shown in Fig. 3(a), our scheme (using `unsync_all`) shows consistent improvement of the baseline SiT model, and significant improvement over the reference DiT model. Second, as shown in Fig. 3(b), the different settings of time step unsynchronization behave differently. Overall, the finest split by `unsync_all` obtains the best performances consistently, followed by `unsync_vec` and `unsync_patch` which split along feature and spatial dimensions and have almost indistinguishable performances. The worst performance is obtained by `unsync_none`, *i.e.* the setting using no

---

[2]This batch mixing scheme may be suboptimal. In preliminary tests (A.8) we found that blending the split timesteps with synchronized ones gives even better results. Searching the optimal scheme is left for future work.

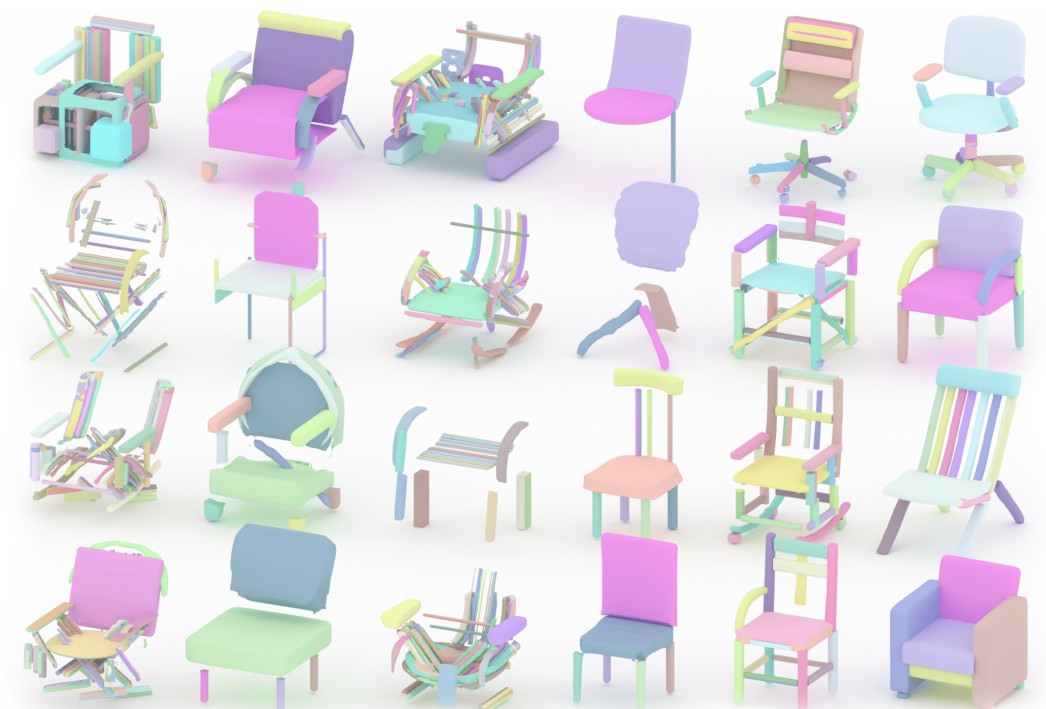

unsync_none  unsync_part  unsync_att  unsync_att_part  unsync_vec  unsync_all

Figure 5: **Results of structured shape generation by different settings**. Semantic parts are colored randomly. Settings exploiting stronger combinatorial stochasticity show better results. In comparison, `unsync_none` that does not apply *ComboStoc* nearly fails to generate meaningful shapes.

combinatorial stochasticity. Fig. 4 visualizes the results of different settings along training steps, where we see better structured images emerge earlier for settings using stronger combinatorial complexity. The comparison among these four settings shows that fully utilizing the combinatorial complexity indeed helps network training.

Note that due to the differences introduced in the timestep embedding module, `unsync_none` has slightly worse performance than the baseline SiT, probably because the timestep encoding vector has a smaller size (see Appendix A.3). While it may be possible to align `unsync_none` with baseline SiT by introducing more capable embedding layers, `unsync_all` already outperforms the baseline with significant margins (Fig. 3(a)). In addition to the result quality, in Appendix A.7 we provide detailed analysis of the computational complexity of our model in comparison with baseline SiT and DiT models, and find that our model is as efficient as the baselines.

**Structured 3D shapes**   We show that for the task of structured 3D shape generation, which has even stronger combinatorial complexity due to the flexible parts and their multiple attributes, our scheme becomes more important to the extent of being indispensable.

For structured 3D shape generation, we identify combinatorial complexity in the following axes: attributes/feature vectors, and spatial parts. Therefore, we obtain $3\times2 = 6$ settings, *i.e.*, `unsync_-none` and `unsync_part` which apply the same or different time schedules to parts respectively, `unsync_att` and `unsync_att_part` which use attribute level schedules, and `unsync_vec` and `unsync_all` which use the most finely divided feature vector level schedules. See Tab. 2(b) for a summary of the 6 configurations. Because of the relatively small size of the PartNet dataset (18K shapes in total, mostly in *chair* and *table* classes), we deem it easier to learn and simply apply the corresponding asynchronous time steps to all samples in each batch, in contrast to the mixing scheme of ImageNet training. We report results at 1.5K epochs, since earlier results cannot be decoded into valid manifold shapes for evaluation in settings like `unsync_none`. In Appendix A.4-A.6 we provide additional network details for structured 3D shape generation.

As shown in Fig. 5, the more combinatorial complexity we exploit, the better performance of the trained network. In comparison, the baseline setting without combinatorial stochasticity, `unsync_-`

| | unsync_none | unsync_part | unsync_att | unsync_att_part | unsync_vec | unsync_all |
|---|---|---|---|---|---|---|
| **FPD**↓ | 7.99 | 4.71 | 7.47 | **3.51** | 4.62 | **4.04** |
| **COV**↓ | 1.32 | 1.03 | 1.83 | **0.85** | 0.97 | **0.86** |
| **MMD**↓ | 1.23 | 1.95 | 1.38 | 1.04 | **0.63** | 0.68 |

Table 3: **Quantitative evaluation of structured shape generation by different settings**. Chair category is used. **Best** scores are marked in bold and underlined; **second best** scores in bold.

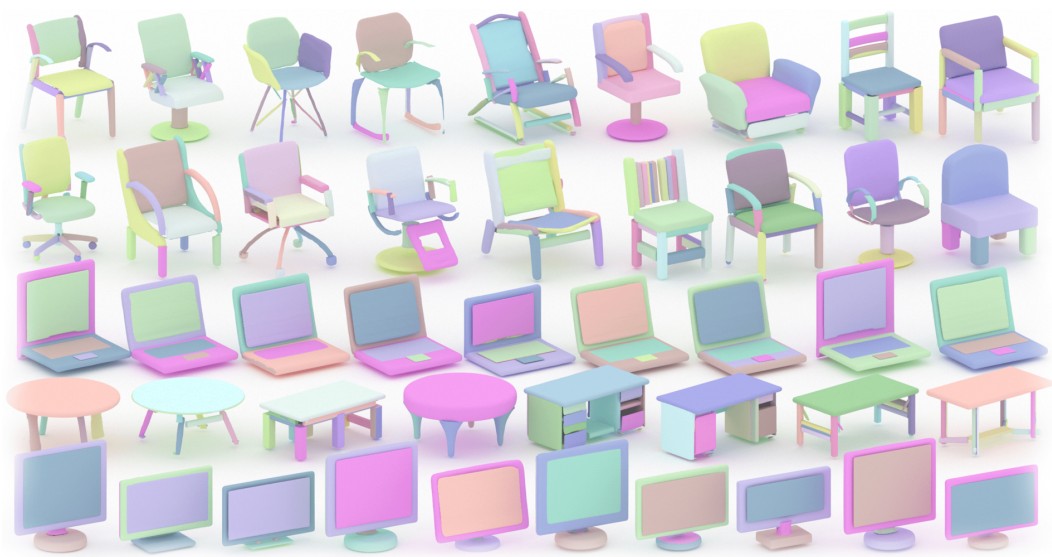

Figure 6: **Class-conditioned generation of structured 3D shapes.** From top to bottom the classes are: chair, laptop, table and display.

none, almost entirely fails to produce meaningful shapes. Moreover, since this task models the highly flexible composition of various numbers of parts, applying the spatial part unsynchronization (Tab. 2(b)) helps obviously, as shown through the three pairs of columns in Fig. 5 (*e.g.*, part vs none, att_part vs att, and all vs vec.).

We report quantitative results in Tab. 3 using the chair category. Following Wang et al. (2023) we use three metrics, including Frechet Point Distance (FPD) that measures the FID on sampled point clouds, coverage (COV) that measures how well each GT sample is covered by the closest generated sample, and minimum matching distance (MMD) that measures how well each generated sample resembles the closest GT sample. The numerical results again show that the part level combinatorial stochasticity enhances generative performance significantly, and unsync_all shows the best overall result.

In Fig. 6 and Appendix A.8 we show more random samples generated by the unsync_all setting. In Appendix A.8, we also compare with other works that generate structured shapes by taking a hierarchical refinement process and find that our results are within their performance ranges.

### 4.2 APPLICATIONS ENABLED BY COMBINATORIAL STOCHASTIC PROCESS

The asynchronous timesteps for different dimensions and attributes of *ComboStoc* enables a novel test time application, namely the chance to specify different degrees of preservation of a data sample to its dimensions and attributes. Specifically, given $\mathbf{t}_0$ specifying the weights in $[0, 1]$ to preserve the data of $\mathbf{x}$, we sample the generative process starting from

$$\mathbf{x}_0 = (1 - \mathbf{t}_0) \odot \mathbf{z} + \mathbf{t}_0 \odot \mathbf{x}, \qquad (3)$$

and increase the time steps for individual dimensions and attributes via $\frac{1-\mathbf{t}_0}{N}$ for $N$ steps. Examples of such asynchronous generative processes are shown in Figs. 7, 8 for images, and Figs. 9, 10 for structured 3D shapes.

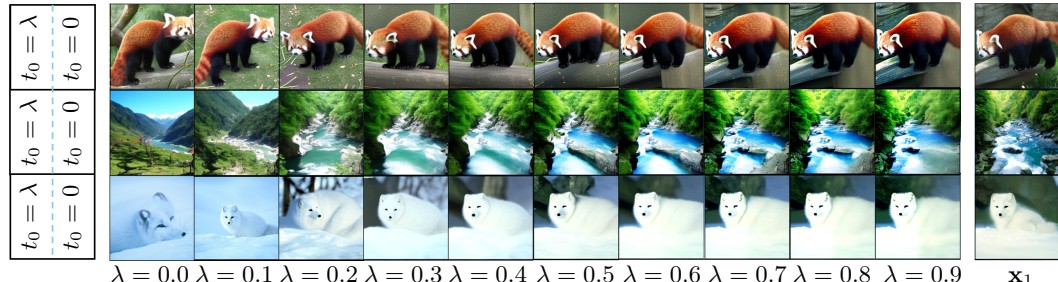

Figure 7: **Image generation using different weights of preservation.** Each reference image (right) is split into two vertical halves (left), and the left half is given the preservation weights while the right region starts from scratch.

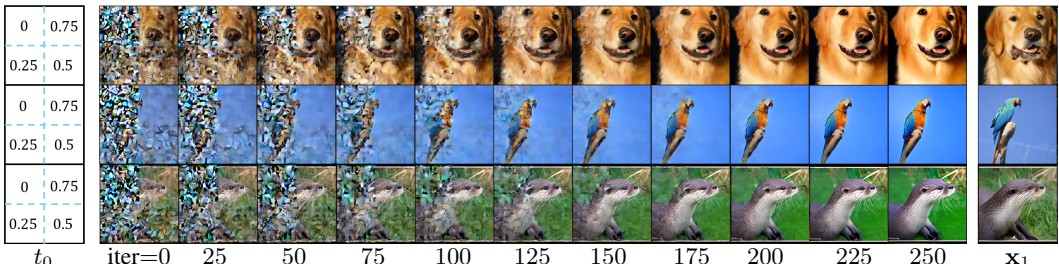

Figure 8: **Image generation with spatially different preservation weights.** As shown in the left column, the four quadrants use $t_0 = 0, 0.25, 0.5, 0.75$, respectively. The sampling iterations converge to results that preserve the corresponding quadrants from the reference images (right) differently.

**Images** In Fig. 7 we show that giving different $t_0$ to a half of a reference image while leaving the other half to generate from scratch, we can achieve different degrees of preservation of the reference images. In particular, as the preservation weight increases from 0 to 1, the preservation of reference content is strengthened. We note that at 0.5 the weight is good enough to preserve most of the reference content. In Fig. 8, we use different preservation weights encoded by $t_0$ for the four quadrants of each image, and show intermediate results along the iterative SDE integration process. From the three examples we can see that stronger weights cause better preservation of reference regions, and the different regions are filled with coherent content despite the spatially varying time schedules. This mode of controlled generation is novel, compared with the binary inpainting mode proposed for standard diffusion models (Lugmayr et al., 2022), where regions of an image are divided into two discrete types, *i.e.* those to preserve and those to generate from scratch.

In Appendix A.8 we show more cases of graded control over image generation. In particular, we find that channel-varying $t_0$ reveals interesting observations about the different contents of latent image encoding (Rombach et al., 2022).

**Structured 3D shapes** By controlling different parts and attributes of structured 3D shapes, we can achieve diverse effects, including shape completion and part assembly. In Fig. 9, we fix the bases of chairs by giving them $t_0 = 0.9$, and complete them with meaningful but diverse structures that satisfy the class condition. The given bases have the chance of being slightly updated to adapt to the completed shapes. In Fig. 10, we randomly position a set of parts, and let the network arrange them into proper shapes, by giving the part shape codes $\mathbf{e}$ and bounding box sizes large preservation weights ($t_0 = 0.9$) and making the rest attributes free to be generated. Here we have considered a simplified setting where the part rotations are given, and leave the more challenging case of rotating shape parts as future work.

## 5 RELATED WORKS

For image generation, while there are numerous optimizations of training schemes, including training loss weights and time schedules (Hang et al., 2023), speedup by distillation (Meng et al., 2023), and sampling path consistency (Song et al., 2023), few have noticed the factor of combinatorial

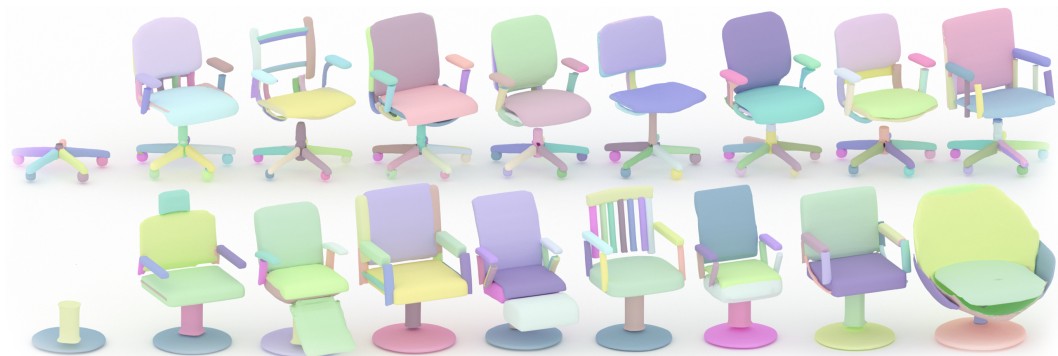

Figure 9: **Structured shape completion.** Given base parts (left), the network can complete the missing parts conditioned on a shape category name (`chair` in this example). While the completed parts show great diversity, the given parts are preserved faithfully.

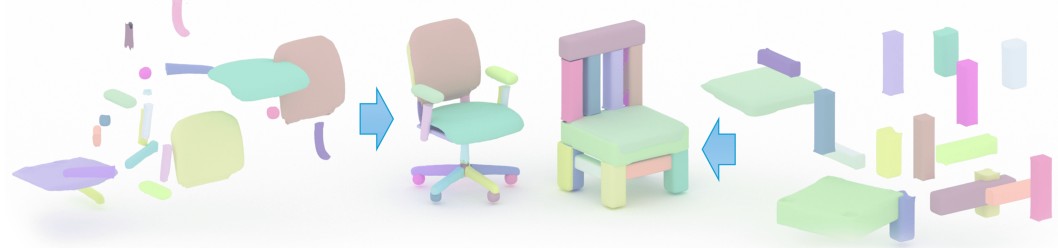

Figure 10: **Assembly of semantic parts.** Given parts in random positions (left and right), the network assembles them into complete shapes (middle). We solve this part-assembly problem via preserving the attributes of part shapes and scales and only generating the attribute of part positions.

complexity. An exception is Gao et al. (2023), which attributes the slow training of DDPM based DiT model (Peebles & Xie, 2023) to the pixel-wise regression loss, which does not emphasize the structural correlation of different patches sufficiently. To address the problem, Gao et al. (2023) design a mask-and-diffusion scheme that masks out portions of the input diffused images during training to encourage learning the patch correlation, implemented by a complex encoder-decoder network with additional side-interpolation modules. In comparison, our scheme is simple and requires minimum changes of baseline networks, but can already improve training of SiT models significantly. Notably, SiT models already surpass DiT models in performance (Ma et al., 2024).

While 3D diffusion generative models are increasing (Zheng et al., 2023; Zhang et al., 2023), few works have been done for structured shape generation. Mo et al. (2019a) studies the representation learning of hierarchically structured shapes and proposes to generate variations using a VAE model. Compared to Mo et al. (2019a), Wang et al. (2023) proposes a rewriting model to enable generalizable cross-category generation. In comparison, we focus on generating flatly structured 3D shapes with leaf level semantic parts. Moreover, by specifying parts and attributes independently, our model enables diverse tasks like shape completion and assembly. Previously, such diverse applications have been studied individually via specialized solutions (Huang et al., 2020; Sung et al., 2015), but in this paper we have shown that they can be potentially unified by a single model generating highly structured data.

# 6    CONCLUSION

We have proposed to focus on the problem of combinatorial complexity of high-dimensional and multi-attribute data samples for diffusion generative models. In particular, we note that for one-sided stochastic interpolants that model many variants of diffusion and flow based models, there exists the problem of under-sampling regions of the path space where the dimensions/attributes are off-diagonal or asynchronous. We propose to fix this issue by sampling the whole space spanned by combinatorial complexity uniformly. Experiments across two data modalities show that indeed by utilizing the combinatorial complexity, performances can be enhanced, and new generation paradigms can be enabled where different attributes of a data sample are generated in asynchronous schedules to achieve

varying degrees of control simultaneously. We hope that our work can inspire future works that look through the combinatorial perspective of generative models.

**Limitation and future work**   Our *ComboStoc* scheme will only have significant effects when the data has combinatorial and structural information, such as different patches for images and different parts for 3D shapes. When the data resides in a vector space whose dimensions are nearly independent, it is hard to exploit the correlation of dimensions and train a model that works well under the combinatorial schedule of different dimensions. Indeed, in such a case the individual dimensions may ideally be generated separately by different models. However, we note that many data types in real life contain strong structural and combinatorial information; particularly eminent are tasks within scientific domains, including molecule docking and protein folding (Corso et al., 2022; Wu et al., 2024; Yim et al., 2024), where diffusion models that better handle their combinatorial structures can be desirable.

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

# A APPENDIX

## A.1 SAMPLING DENSITIES FOR TYPICAL DIFFUSION AND *ComboStoc* DIFFUSION

We show that the sampling density for a typical linear interpolant diffusion process is different from that of the *ComboStoc* diffusion process. In particular, we show that the typical diffusion creates nonuniform density making regions away from target data points low in probability. On the other hand, the *ComboStoc* diffusion creates uniform densities for subspaces spanned by source and target data points. Therefore, *ComboStoc* diffusion improved sampling.

Without loss of generality, suppose the subspace $\mathcal{R}$ spanned by $\mathbf{z}$ and $\mathbf{x_1}$ has $\mathbf{z}$ as the minimum corner, and $\mathbf{x_1}$ as the maximum corner. Thus we have $\mathcal{R} = \{\mathbf{x} | \mathbf{z} \preceq \mathbf{x} \preceq \mathbf{x_1}\}$.

First, we note that for ComboStoc diffusion, any point $\mathbf{x} \in \mathcal{R}$ has a uniform density by definition (Fig. 2(b)).

Second, we show that the typical diffusion creates nonuniform density. As shown in Fig. 2(a), we denote an arbitrary point $\mathbf{x} \in \mathcal{R}$, and a moving point $\mathbf{p}_t = (1-t)\mathbf{z} + t\mathbf{x}_1$ along the diagonal connecting $\mathbf{z}$ and $\mathbf{x_1}$.

We denote by $\rho(\mathbf{x})$ the probability density for sampling $\mathbf{x}$, generated by any Gaussian distribution $G_{\mathbf{p}_t} = N(\mathbf{p}_t; (1-t)^2 \mathbf{1})$ centered at $\mathbf{p}_t$ and with variance scaled by the interpolation coefficient $1 - t$. Therefore, we have

$$\rho(\mathbf{x}) = \int_0^1 G_{\mathbf{p}_t}(\mathbf{x})dt = \int_0^1 \frac{1}{\sqrt{2\pi}(1-t)} e^{-\frac{\|\mathbf{x}-\mathbf{p}_t\|^2}{2(1-t)^2}} dt. \tag{4}$$

Plugging in the parameterized equation of $\mathbf{p}_t$, we have

$$\rho(\mathbf{x}) = \frac{1}{\sqrt{2\pi}} \int_0^1 \frac{1}{1-t} e^{-\frac{\|t(\mathbf{z}-\mathbf{x}_1)+\mathbf{x}-\mathbf{z}\|^2}{2(1-t)^2}} dt. \tag{5}$$

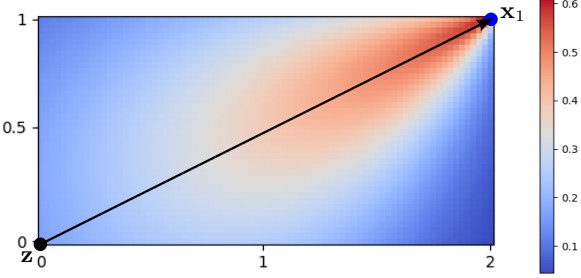

Figure 11: Visualizing the sampling density $\rho(\mathbf{x})$ of a typical one-sided linear interpolant (see Fig. 2(a)) by numerical integration. One can see the obvious tendency of shrinking coverage toward the target data point.

The above integration does not have closed-form solution, so we visualize $\rho$ by numerical integration in Fig. 11, where we see the clear tendency of shrinking coverage toward data points. Moreover, we find the gradient $\nabla \rho$ is more friendly to work with. In particular, by straightforward calculation we have

$$(\mathbf{x}_1 - \mathbf{x}) \cdot \nabla \rho(\mathbf{x}) = \frac{1}{\sqrt{2\pi}} e^{-\frac{\|\mathbf{x}-\mathbf{z}\|^2}{2}} > 0, \tag{6}$$

which means that $\nabla \rho(\mathbf{x})$ has a positive projection along the direction $\mathbf{x}_1 - \mathbf{x}$. Therefore, we can conclude that the sampling density $\rho(\mathbf{x})$ is not uniform and grows when approaching the target data points.

## A.2 MINIMIZING THE OFF-DIAGONAL DRIFT

The problem of off-diagonal drift can be revealed from the perspective of test-time integration. For ease of analysis we assume an ODE is solved to follow the prescribed velocity field to move from the source noise point to the target data point.

If the integration never stumbles on sample points off diagonal line connecting $\mathbf{z}$ and $\mathbf{x}_1$, the integration starting from a point $\mathbf{x}_{t_0} = \mathbf{z} + t_0(\mathbf{x}_1 - \mathbf{z})$ and following the velocity field $\mathbf{x}_1 - \mathbf{z}$ would always end at the target point, i.e.,

$$\mathbf{x}_{t_0} + \int_{t_0}^{1} (\mathbf{x}_1 - \mathbf{z})dt = \mathbf{z} + t_0(\mathbf{x}_1 - \mathbf{z}) + (1 - t_0)(\mathbf{x}_1 - \mathbf{z}) = \mathbf{x}_1. \tag{7}$$

Now suppose the integration stumbles upon an off-diagonal sample point $\mathbf{x}_{\mathbf{t}_0}$ at a tensorized interpolation schedule $\mathbf{t}_0$ with different values for its various entries. The integration by following only the velocity $\mathbf{x}_1 - \mathbf{z}$ would miss the target data point $\mathbf{x}_1$ (Fig. 2(c), dotted arrow). Precisely,

$$\mathbf{x}_{t_0} + \int_{t_0}^{1} (\mathbf{x}_1 - \mathbf{z})dt \tag{8}$$
$$= \mathbf{z} + \mathbf{t}_0 \odot (\mathbf{x}_1 - \mathbf{z}) + (1 - t_0)(\mathbf{x}_1 - \mathbf{z})$$
$$= \mathbf{x}_1 + (\mathbf{t}_0 - t_0) \odot (\mathbf{x}_1 - \mathbf{z}),$$

where $t_0 = \min(\mathbf{t}_0)$ denotes the minimum interpolation schedule across the dimensions of $\mathbf{t}_0$. Here we assume the test-time setting that for asynchronous timesteps we integrate until the slowest one finishes.

To address this divergence problem, we propose two possible approaches for mitigation: 1) we can minimize the off-diagonal drift by following the negative gradient of a drift potential, and 2) we can design a cone-shaped velocity field, such that the integration converges to target data points.

**Off-diagonal drift minimization** The off-diagonal offset vector $\delta(\mathbf{x}_t)$ can be defined as

$$\delta(\mathbf{x}_t) = -\mathbf{v}_{cmpn} = \mathbf{x_t} - \mathbf{x}_1 - \frac{(\mathbf{x_t} - \mathbf{x}_1) \cdot (\mathbf{x}_1 - \mathbf{z})}{||\mathbf{x}_1 - \mathbf{z}||^2}(\mathbf{x}_1 - \mathbf{z}). \tag{9}$$

To minimize the drift, we can simply follow its negation $\mathbf{v}_{cmpn}$ in addition to the original velocity during integration, which is equivalent to minimizing a drift potential $\Phi\left(\delta(\mathbf{x_t})\right) = \frac{1}{2}\|\delta_\mathbf{t}\|^2$ by gradient descent, and promotes the convergence to target data points (Fig. 2(c), dashed arrow).

**Cone-shaped velocity field** Different from the off-diagonal drift minimization, we can also design a cone-shaped velocity field that generalizes the simple constant velocity $\mathbf{x}_1 - \mathbf{z}$ to a cone of velocities covering the expanded region $\mathcal{R}$. In particular, we can use the following velocity

$$\mathbf{v}_{\mathbf{t}_0} = \frac{\mathbf{x}_1 - \mathbf{x}_{\mathbf{t}_0}}{1 - t_0}, \tag{10}$$

where again $t_0 = \min(\mathbf{t}_0)$ denotes the minimum interpolation schedule across the dimensions of $\mathbf{t}_0$. Note that for synchronized schedule, $\mathbf{v} = \frac{\mathbf{x}_1 - \mathbf{x}_{t_0}}{1 - t_0} = \frac{\mathbf{x}_1 - \mathbf{z} - t_0(\mathbf{x}_1 - \mathbf{z})}{1 - t_0} = \mathbf{x}_1 - \mathbf{z}$, i.e., the original velocity is a special case of this velocity field. To see why this is a cone shaped velocity field, note that for a timestep $\mathbf{t}_\lambda = \lambda\mathbf{t}_0 + (1 - \lambda)\mathbf{1}$ along the line of $\mathbf{t}_0$ and $\mathbf{1}$, their velocities are equal. Therefore, it is easy to see such a constant velocity along the line connecting an off-diagonal point and the target point would lead to convergence to the target data point. Precisely,

$$\mathbf{x}_{\mathbf{t}_0} + \int_{t_0}^{1} \frac{\mathbf{x}_1 - \mathbf{x}_{\mathbf{t}_0}}{1 - t_0}dt = \mathbf{x}_{\mathbf{t}_0} + \mathbf{x}_1 - \mathbf{x}_{\mathbf{t}_0} = \mathbf{x}_1, \tag{11}$$

due to the cone-shaped velocity field.

Throughout our experiments, we have used the first approach of off-diagonal drift minimization to mitigate the divergence issue, and leave the test of the second approach for future work due to limited computational resources.

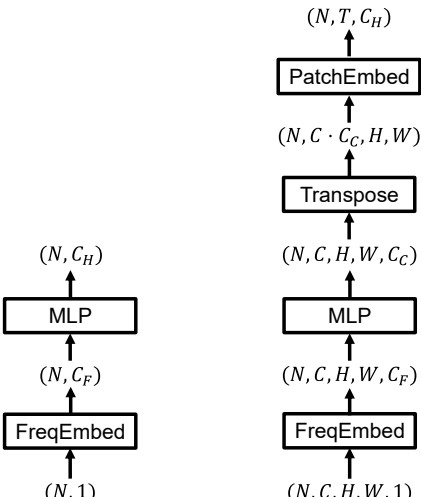

Figure 12: **Adaptation of time step embedding for *ComboStoc*. Left**: the original time step embedding module of SiT, where $N$ is batch size, $C_F$ is the sine/cosine frequency embedding length, and $C_H$ is the hidden dimension of SiT transformer. **Right**: the adapted time step embedding module for *ComboStoc*. $\mathbf{t}$ is now of shape $(N, C, H, W)$. The first two layers remain the same as the original module, applying to each entry of tensor $\mathbf{t}$ and producing a compressed time step encoding of dim $C_C$. Given the result tensor of shape $(N, C, H, W, C_C)$, we further transpose it to combine the channel dimensions and use the same patchwise embedding layer as SiT (and ViT) to embed the local patches into vectors of dim $C_H$. Suppose the patch size is $L \times L$, then $T = H \times W / L^2$.

### A.3   SiT ADAPTATION FOR IMAGE GENERATION

As shown in Fig. 12, given a tensorized time step $\mathbf{t}$ of shape $(N, C, H, W)$ that is the same as the latent encoding of input images, we not only encode each of the time steps for different dimensions as done before, but also embed the result feature map of time steps in the same way as image embedding, *i.e.* the patch-wise embedding originally from ViT (Dosovitskiy et al., 2021). This design ensures that the different dimensions are conditioned on their corresponding time steps in addition to the shared class label. Note that to avoid introducing large embedding layers, we have used $C_C = 4$ to encode a timestep scalar, which is significantly smaller than the $C_H$ of SiT. This can be the reason why `unsync_none` performs slightly worse than the baseline SiT (Sec. 4), both of which have exactly the same network elsewhere.

### A.4   ENCODING FOR STRUCTURED 3D SHAPE GENERATION

We have adopted the pretrained part shape encoding network from Wang et al. (2023). In particular, Wang et al. (2023) design a point cloud VAE to encode 3D shapes into a sparse set of latent codes, and on top of the latent set, they train another transformer VAE to compress them into a single latent code. Therefore, each part shape from the PartNet dataset (Mo et al., 2019b) is normalized into unit size and encoded into a single code, which allows us to represent structured 3D shapes as a collection of parts as detailed in Sec. 3.

The embedding modules for part existence and bounding box follow the same design as timestep embedding. That is, we first turn each of the scalar dimensions into frequency codes using the sine/cosine embedding, and then embed them into vectors of dim 4 (*cf.* Fig. 12), before finally embedding each of the collective attributes as a whole into vectors of hidden dim 384, through respective FC layers.

(a) Image generation

| Setting | $\mathbf{t}$ |
|---------|--------------|
| unsync_none | $(N, 1, 1, 1)$ |
| unsync_patch | $(N, 1, H, W)$ |
| unsync_vec | $(N, C, 1, 1)$ |
| unsync_all | $(N, C, H, W)$ |

(b) Structured 3D shapes

| Setting | $\mathbf{t}$ |
|---------|--------------|
| unsync_none | $(N, 1, 1)$ |
| unsync_part | $(N, L, 1)$ |
| unsync_att | $(N, 1, [1, 1, 1])$ |
| unsync_att_part | $(N, L, [1, 1, 1])$ |
| unsync_vec | $(N, 1, V)$ |
| unsync_all | $(N, L, V)$ |

Table 4: **Time step tensor shapes of different configurations**. **Left**: images are of shape $(N, C, H, W)$, where $N$ is batch size, $C$ is channel size, $H$ and $W$ are height and width, respectively. The $\mathbf{t}$ tensors match up with the image tensors through broadcast semantics. **Right**: structured 3D shapes are of shape $(N, L, [V_s, V_b, V_e])$, where $N$ is batch size, $L$ is the number of shape parts, $[V_s, V_b, V_e]$ is the concatenation of three attributes, *i.e.*, $V_s = 1$ indicator of existence, $V_b = 6$ bounding box, $V_e = 512$ part shape code; we denote the three attributes collectively as $V$. $\mathbf{t}$ tensors match up with the shape tensors through broadcast semantics.

## A.5 DETAILS OF TENSORIZED TIME STEPS

In Tab. 4 we give the details of split timestep specifications for all configurations, across images and structured 3D shapes. We rely on the broadcast semantics of Numpy and Pytorch to assign synchronized timesteps to multiple dimensions.

## A.6 IMPLEMENTATION DETAILS

The image generation model is modified from SiT-XL/2, *i.e.*, the large model with 28 layers, 1152 hidden dimension, $2 \times 2$ patch size, and 16 attention heads. We trained the model using the default settings of SiT, with AdamW solver and fixed learning rate $10^{-4}$, and batch size 256, on 4 Nvidia H100 gpus. The training takes 7.5 days for 800K iterations. Evaluating the models uses the SDE integrator with 250 steps. The use of classifier-free guidance (CFG) or not is specified at corresponding results. For comparison with baselines in terms of FID-50K, CFG is not used unless otherwise specified. In the result gallery figures, CFG is used with guidance strength 4.0.

The structured 3D shape generation model uses a network of SiT small model, *i.e.*, the model has 12 layers, 384 hidden dimension, 256 tokens for parts and 6 attention heads. We trained the model using the AdamW solver with a fixed learning rate of $10^{-4}$ and batch size 16. We trained the model on 4 Nvidia A100 gpus, which takes 3 days for 1.5K epochs. Evaluating the models uses iterative sampling with 500 iterations; in each iteration, the predicted part existence is binarized via threshold 0.5 before being diffused back for the next iteration. Class conditional sampling without CFG is always applied.

| Methods | Parameters(M) | Mem. Usage(MB) | Training Speed (steps/sec) | GFlpos | Inference Speed(ms) |
|---------|---------------|----------------|----------------------------|--------|---------------------|
| DiT | 675 | 75580 | 0.17 | 237.34 | 49 |
| SiT | 675 | 76868 | 0.19 | 237.34 | 50 |
| Ours | 673 | 76340 | 0.15 | 352.46 | 48 |

Table 5: Comparison of computational complexity with SiT and DiT, in terms of parameter count, training stage speed and memory usage, and inference stage speed and Gflops. All tests are done on a single Nvidia A100-80G GPU at the XL/2 model configuration with input image of size $256 \times 256$. The GFlops are calculated by DeepSpeed.

## A.7 COMPUTATIONAL COMPLEXITY ANALYSIS

We provide a comparison with SiT and DiT in Tab. 5, in terms of parameter count, training stage speed, memory usage, inference stage speed, and GFlops. All tests are done on a single Nvidia

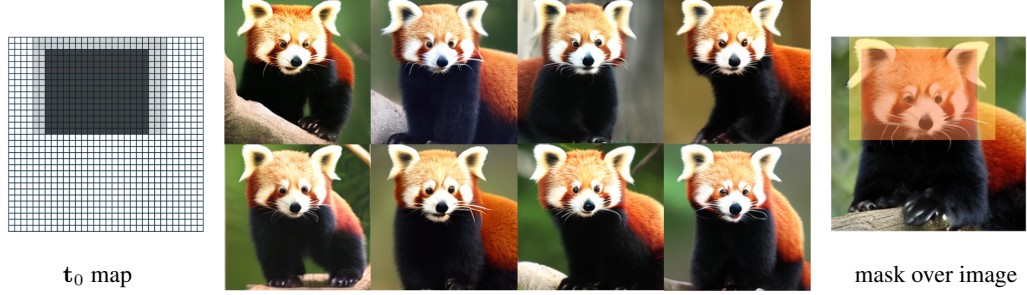

Figure 13: **Spatially and channel varying** $\mathbf{t}_0$. For the spatial dimensions $\mathbf{t}_0[:,:,i,j]$, the assignment is specified in the left column. For the feature channel dimensions $\mathbf{t}_0[:,C,:,:]$, the $C$-dim is given 0.5 and the rest given 0. Therefore, we obtain results that correspond to the reference images ($\mathbf{x}_1$) in complex ways. Notably, earlier channels correspond more to image structures and later channels to image colors.

Figure 14: **Customized graded** $\mathbf{t}_0$. **Left** shows the $\mathbf{t}_0$ map in pixels, where each pixel corresponds to a $8 \times 8$ patch of the original image. The darker region uses $t_0 = 0.75$ and the lighter region uses $t_0 = 0.5$. **Right** overlays the map over the reference image. **Middle** shows generated images.

A100-80G GPU at the XL/2 model configuration, with an input image of size 256×256 and training batch size 256. The GFlops are calculated by DeepSpeed (Rasley et al., 2020).

From Tab. 5 we can see that compared with DiT and SiT, our model has a smaller number of parameters as we use a smaller timestep embedding module (see Appendix A.3). Therefore, our GPU memory cost during training is slightly smaller than SiT. On the other hand, for the conditioning by class label and timestep implemented as a modulation operator (see Fig. 3 of the DiT paper for illustration (Peebles & Xie, 2023)), our conditioning is a tensor of the same shape as the image tensor, in contrast to DiT/SiT's conditioning by a vector only of the channel size of the image tensor; to produce the conditioning tensor involves more computation than the conditioning vector, so our model leads to more flops and slightly increased training cost per step. Nevertheless, the production of the conditioning tensor is a standard MLP feature transformation and fits nicely into GPU parallel computation, so the inference speed is not sacrificed in comparison with DiT/SiT.

### A.8    MORE RESULTS

Fig. 13 shows another example of image generation where we use varying degrees of data preservation across both spatial dimensions (the four quadrants) and feature channel dimensions. In particular, we assign spatial preservation weights according to the left column in the figure, and additionally assign 0.5 to the specified channel index $C$ and 0 to other channels, as shown in the middle four columns. Interestingly, we see that the different channels of the stable-diffusion VAE latent space (Rombach et al., 2022) have very different content. For $C = 0$ the first channel, the generated results mostly preserve the spatial structures of the reference images, and the color cues are largely lost. From $C = 1$ to $C = 3$, the generated results increasingly preserve the color cues of the reference images but lose more of the structures. The findings suggest that earlier channels of the VAE latent space emphasize on structures and later ones on image-level color distributions.

| Category | Method | FPD↓ | COV↓ | MMD↓ |
|----------|--------|------|------|------|
| chair | StructureNet | 4.67 | 0.89 | **0.58** |
|  | StructRe | **2.63** | **0.70** | **0.65** |
|  | Ours | **4.04** | **0.86** | 0.68 |
| table | StructureNet | 6.07 | 1.43 | **0.55** |
|  | StructRe | **1.98** | **0.66** | **0.53** |
|  | Ours | **3.43** | **1.20** | 0.72 |

Table 6: **Comparison on structured 3D shape generation**. Our results are comparable to the baselines that additionally use the hierarchies of shape parts to constrain generations. **Best** scores are marked in bold and underlined; **second best** scores in bold.

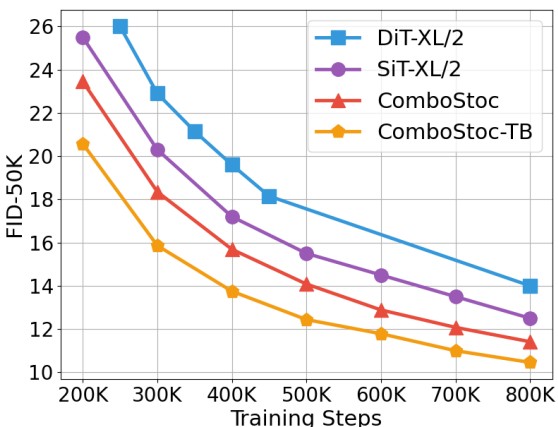

Figure 15: **A different scheme of batch mixing for training image generative model.** Plots include our baseline model (*ComboStoc*) and the new batch mixing model (*ComboStoc*-TB), as well as DiT and SiT for reference; all models are of the scale XL/2 Ma et al. (2024).

Fig. 14 shows an example of using a manually specified mask with graded preservation weights to achieve controlled generation of images. We have assigned a stronger weight ($t_0 = 0.75$) to the face of the red panda, and a lighter weight ($t_0 = 0.5$) to the region around, and let the model generate variations freely at the other regions. As expected, the generated images show different bodies for the same red panda face, with smooth transitions around the face regions.

Tab. 6 gives the comparison between our structured 3D shape generation model and two baselines, *i.e.*, StructRe (Wang et al., 2023) and StructureNet (Mo et al., 2019a), in terms of FPD, COV and MMD. Shapes in PartNet are labeled into semantic parts that are organized into trees, *i.e.*, coarse parts can be decomposed into fine parts by following the tree. Exploiting this hierarchical data, the two baselines expand coarse parts into fine parts progressively, which helps constrain the generated shapes toward better regularity. In comparison, our network does not use this hierarchical information and directly generates the leaf level parts. Nevertheless, the results by `unsync_all` show performances within the baseline results. Visually, we find our results generally show stronger diversity than the shapes by Wang et al. (2023) and Mo et al. (2019a). Finally, it is an interesting topic to study how to combine the approaches of hierarchical generation and diffusion generative models, which have differing advantages in aspects of structure regularity and diversity.

Fig. 15 shows results from preliminary tests on a different scheme of batch mixing for image generative model training (Sec. 4). For training this model named *ComboStoc*-TB, we simply blend the asynchronous time steps and the synchronized ones for a whole batch. In addition, we try to align the time step embedding module with the baseline SiT by setting $C_C = C_H$ (Fig. 12). The two modifications combined lead to even larger improvements over baselines. Indeed, *ComboStoc*-TB speeds up the convergence of SiT by $\approx 1.75\times$. We plan to investigate these modifications thoroughly in the future.

Figs. 16 to 31 show more results generated by *ComboStoc* models for both structured 3D shapes and images.

## A.9 BROADER IMPACT

In this paper we have presented *ComboStoc* which improves and extends baseline diffusion generative models, across tasks of image generation and structured 3D shape generation. Image generation can be misused potentially, although our model as well as the baseline model uses very coarse level class name conditioning that prevents highly targeted applications. Structured 3D shapes are mostly furniture like daily objects, so their generation is unlikely to be misused. In terms of methodology, we have advocated the importance of sampling the combinatorial flexibility for both model performance and new applications. The combinatorial complexity of high-dimensional and multiple-attribute data samples can be further explored theoretically based on our work, for example from the perspective of ergodicity (Walters, 2000) as emphasized by our dense sampling of all possible interpolation points.

972
973
974
975
976
977
978
979
980
981
982
983
984
985
986
987
988
989
990
991
992
993
994
995
996
997
998
999
1000
1001
1002
1003
1004
1005
1006
1007
1008
1009
1010
1011
1012
1013
1014
1015
1016
1017
1018
1019
1020
1021
1022
1023
1024
1025

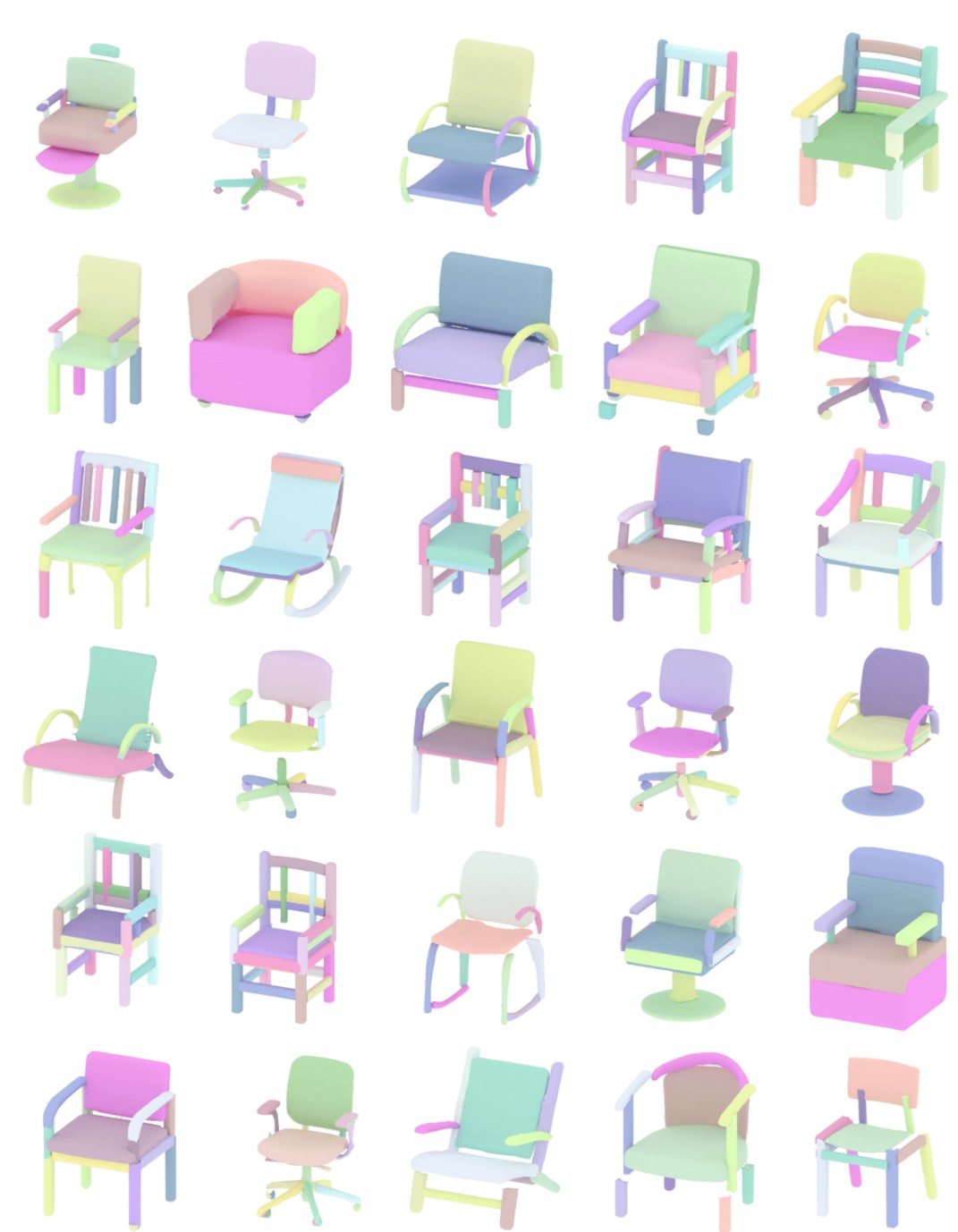

Figure 16: Class-conditioned structured 3D shapes generated by *ComboStoc*. Class label is `chair`.

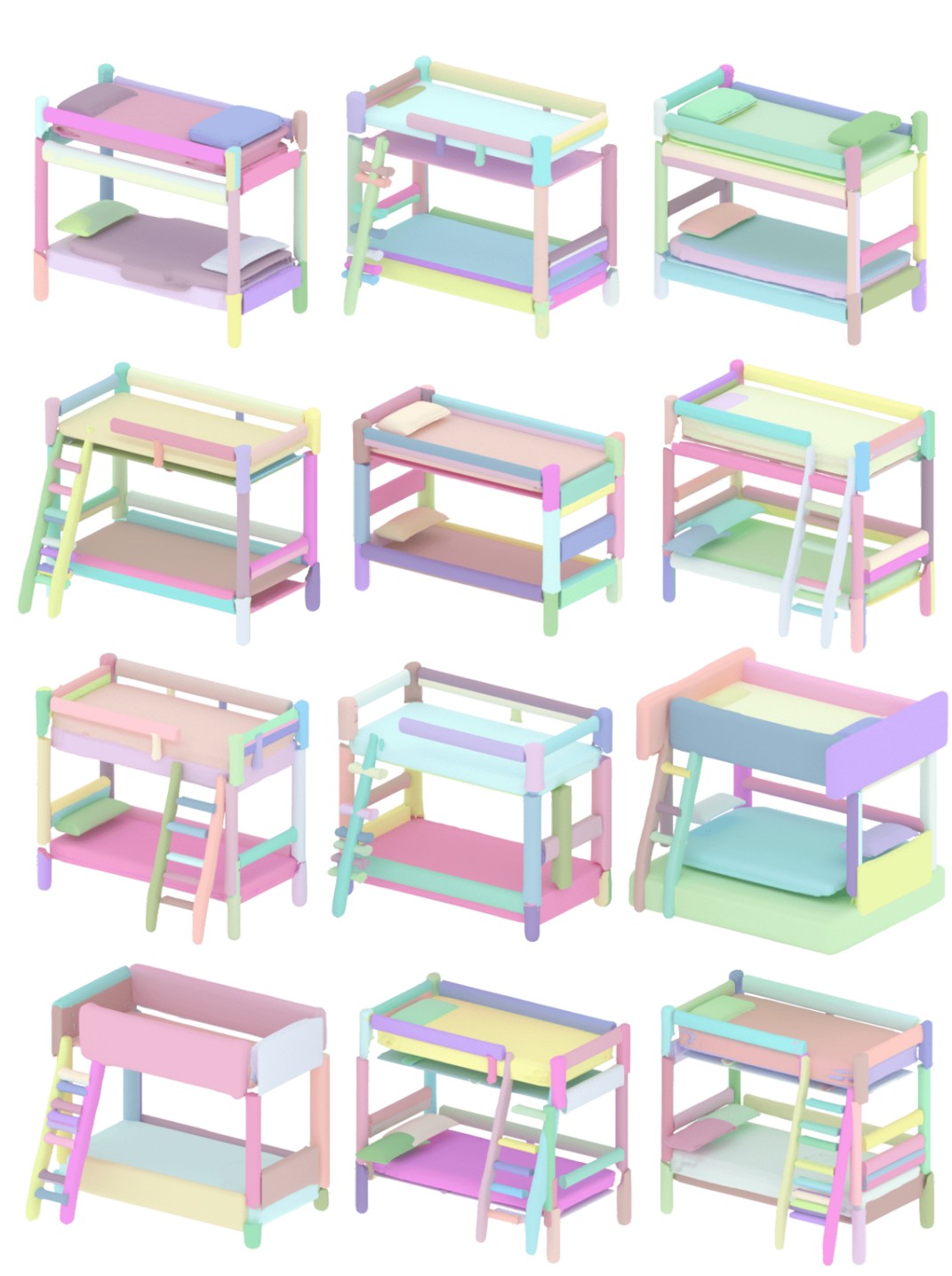

Figure 17: Class-conditioned structured 3D shapes generated by *ComboStoc*. Class label is `bed`.

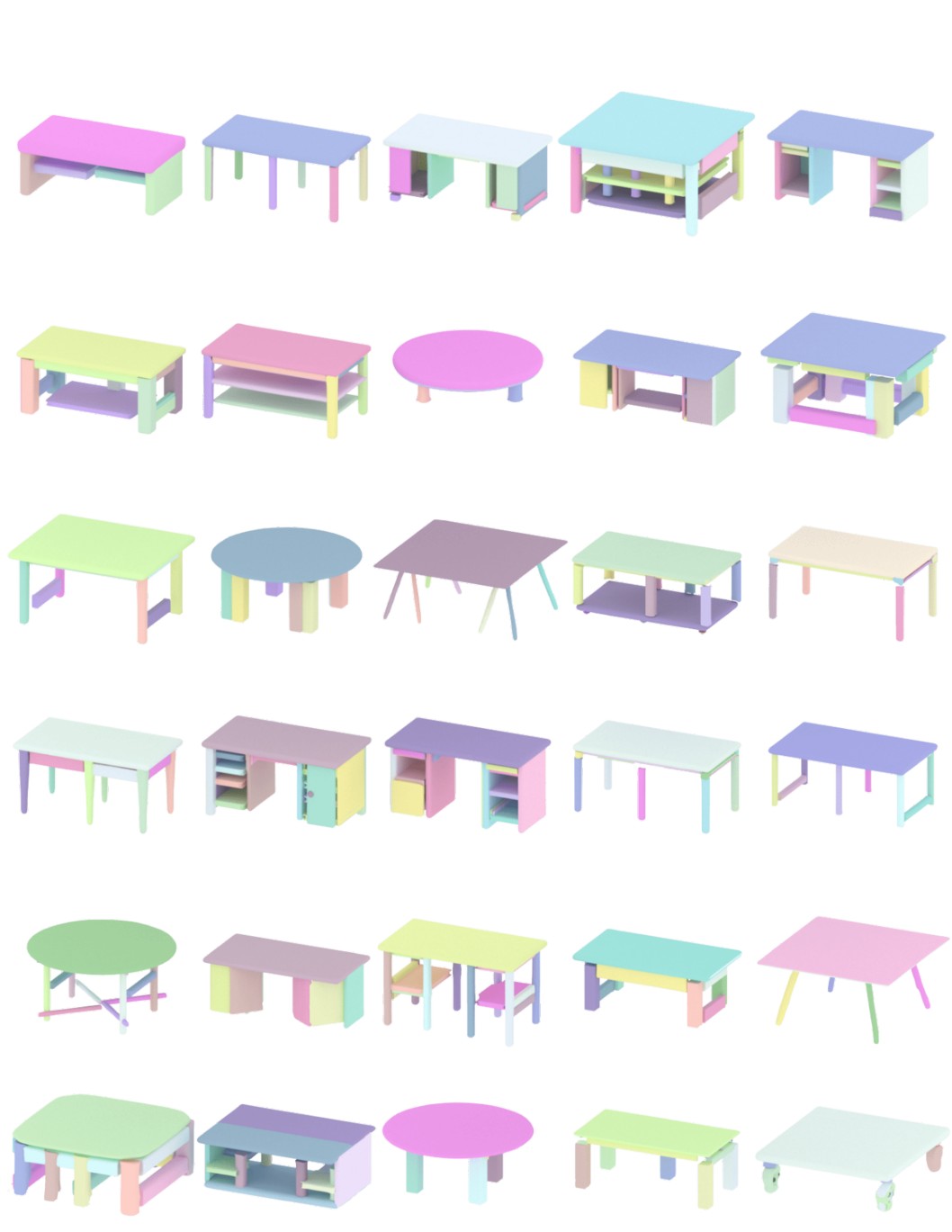

Figure 18: Class-conditioned structured 3D shapes generated by *ComboStoc*. Class label is `table`.

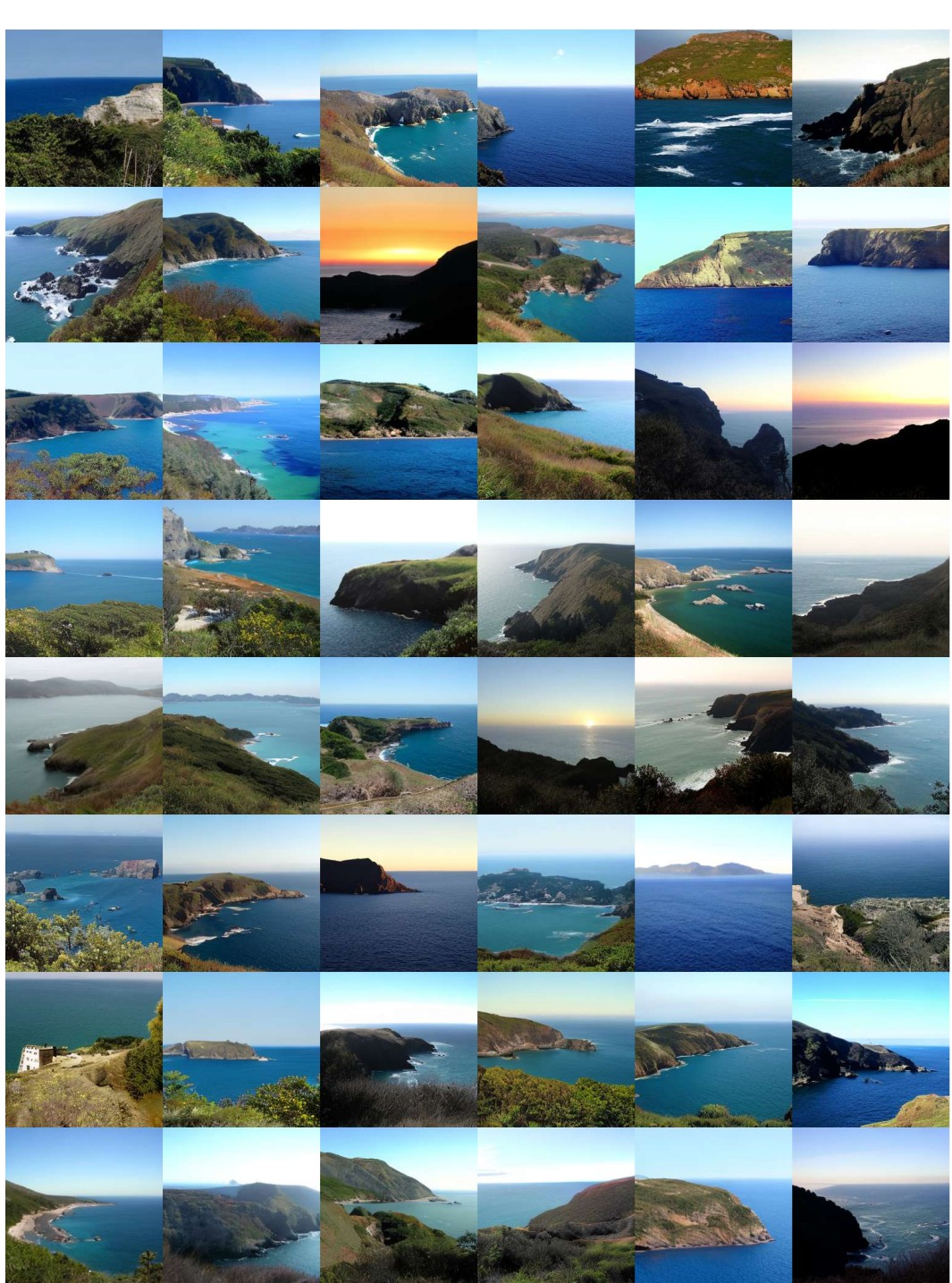

Figure 19: $256 \times 256$ samples of *ComboStoc*-XL/2 800K. Classifier-free guidance scale = 4.0. Class label = "promontory, headland, head, foreland" (976).

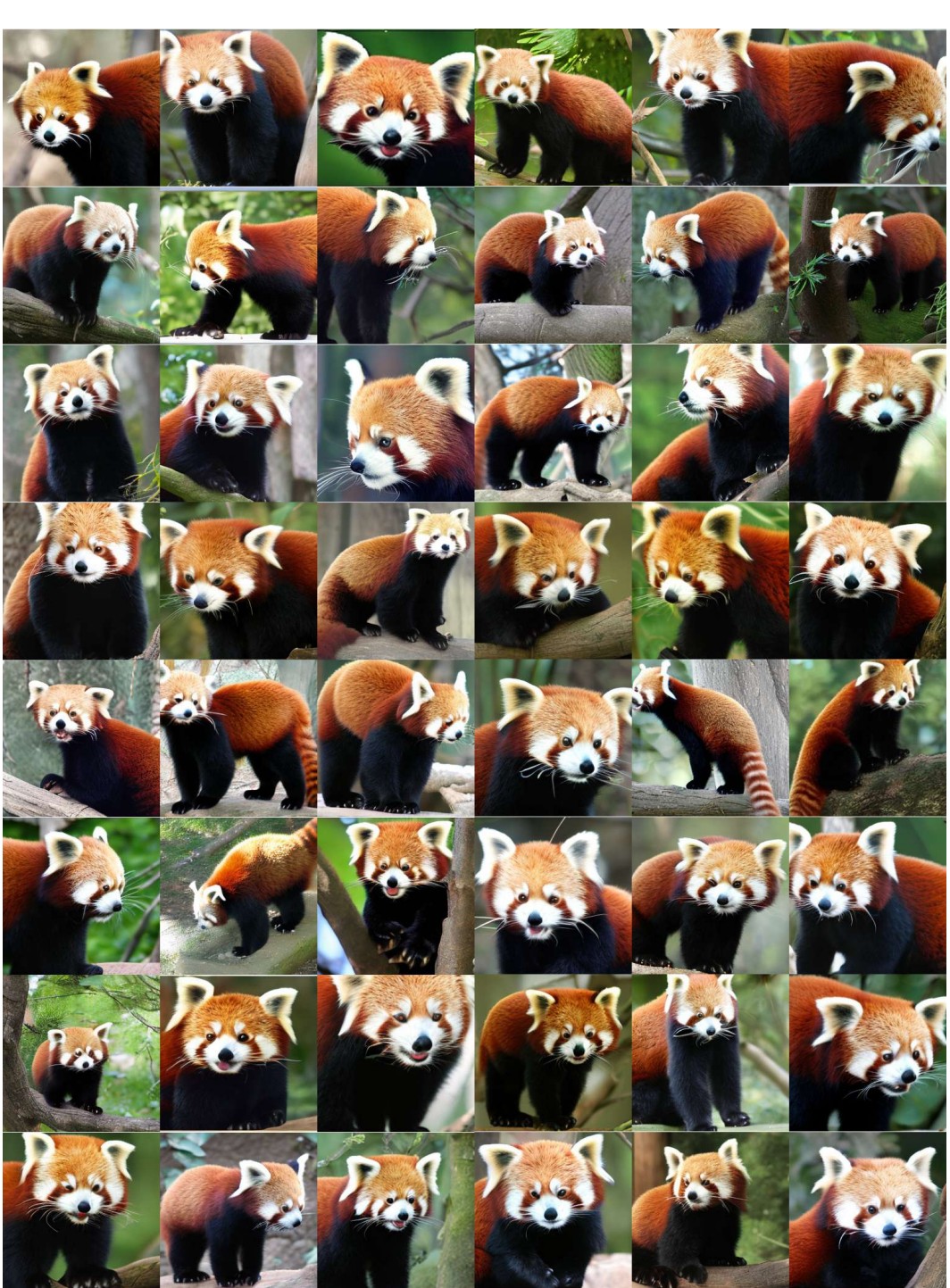

Figure 20: $256 \times 256$ samples of *ComboStoc*-XL/2 800K. Classifier-free guidance scale = 4.0. Class label = "lesser panda, red panda, panda, bear cat, cat bear, Ailurus fulgens" (387).

1242
1243
1244
1245
1246
1247
1248
1249
1250
1251
1252
1253
1254
1255
1256
1257
1258
1259
1260
1261
1262
1263
1264
1265
1266
1267
1268
1269
1270
1271
1272
1273
1274
1275
1276
1277
1278
1279
1280
1281
1282
1283
1284
1285
1286
1287
1288
1289
1290
1291
1292
1293
1294
1295

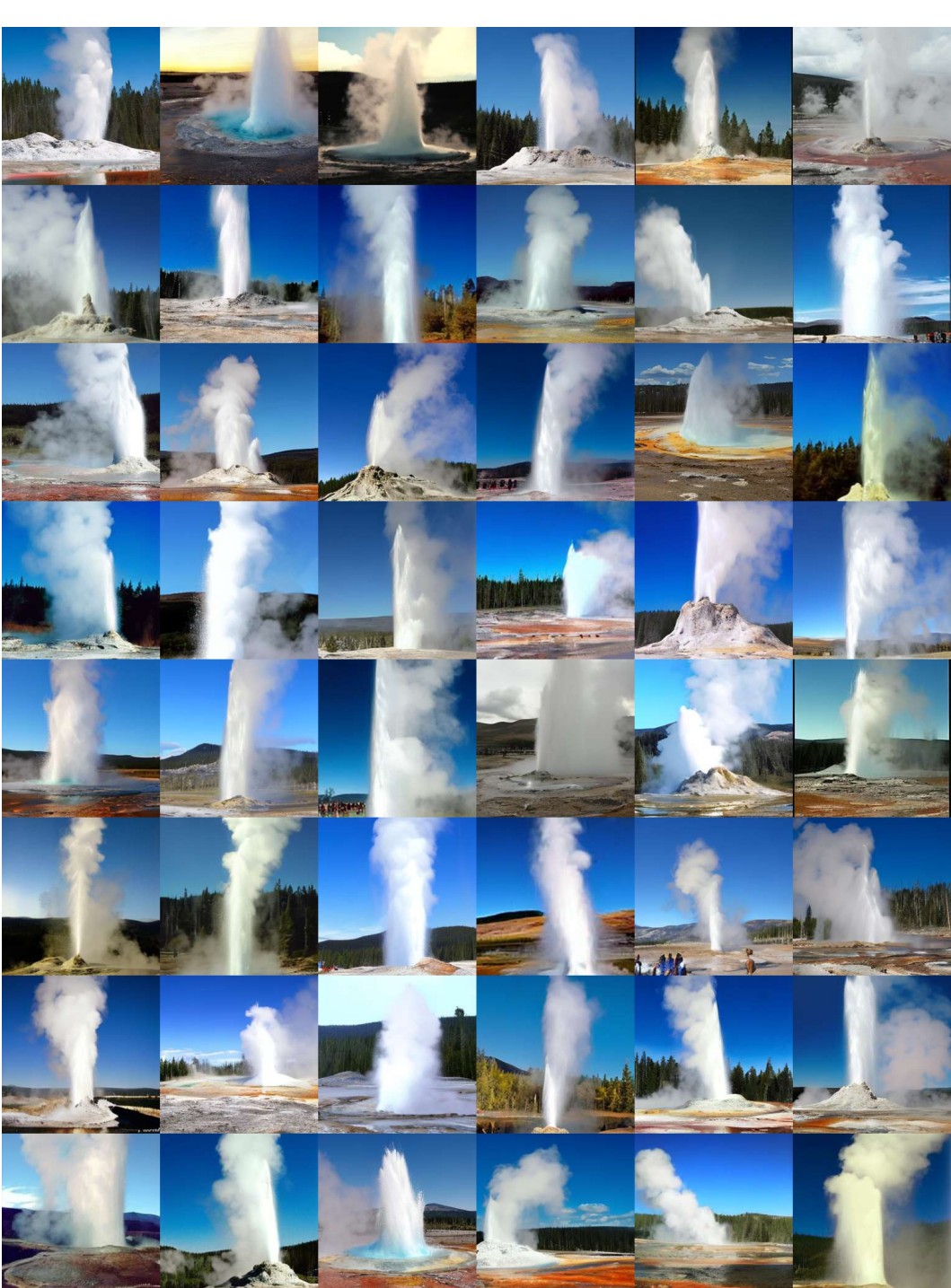

Figure 21: 256 × 256 samples of *ComboStoc*-XL/2 800K. Classifier-free guidance scale = 4.0. Class label = "geyser" (974).

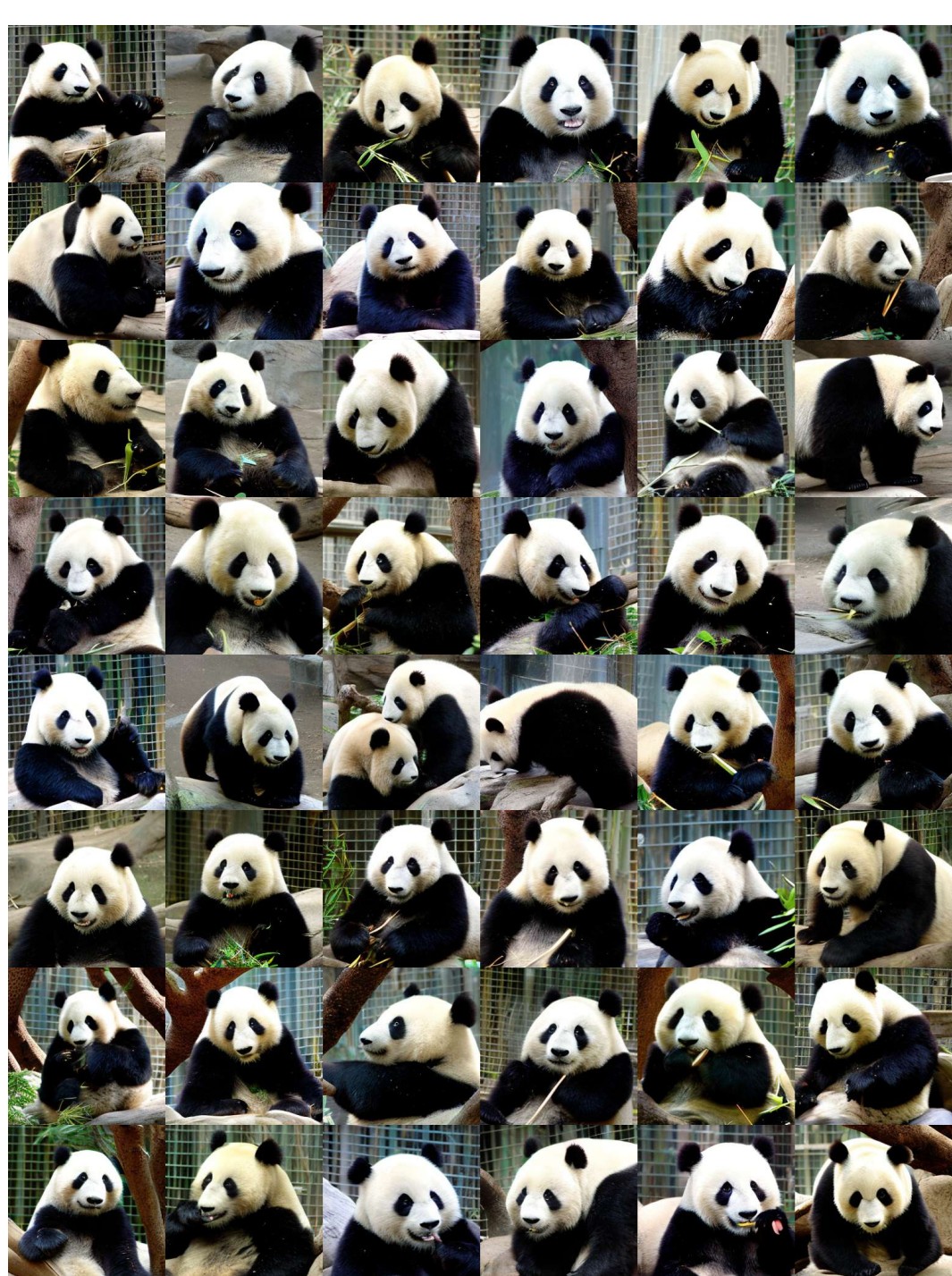

Figure 22: 256 × 256 samples of *ComboStoc*-XL/2 800K. Classifier-free guidance scale = 4.0. Class label = "giant panda, panda, panda bear, coon bear, Ailuropoda melanoleuca" (388).

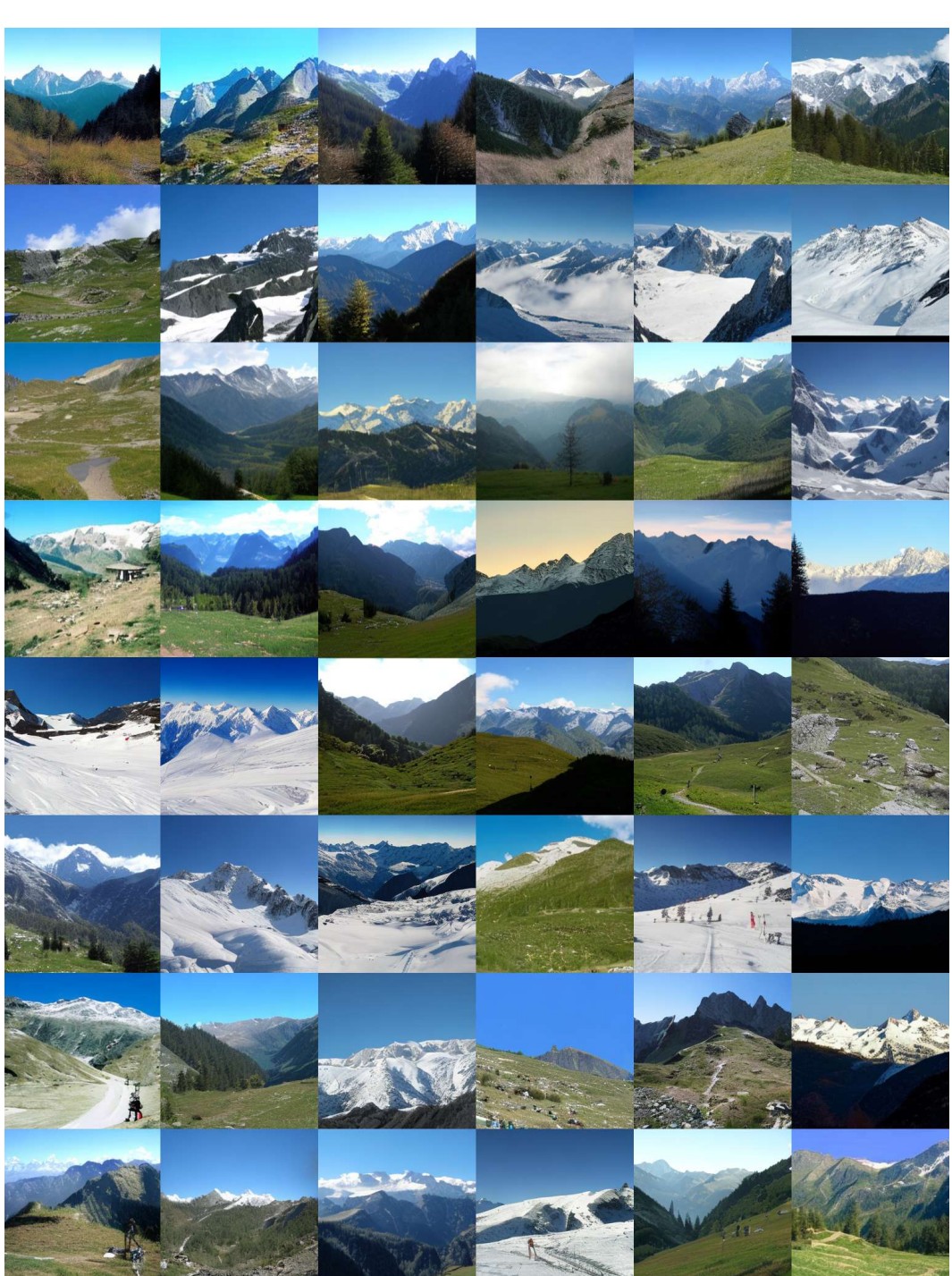

Figure 23: 256 × 256 samples of *ComboStoc*-XL/2 800K. Classifier-free guidance scale = 4.0. Class label = "alp" (970).

1404
1405
1406
1407
1408
1409
1410
1411
1412
1413
1414
1415
1416
1417
1418
1419
1420
1421
1422
1423
1424
1425
1426
1427
1428
1429
1430
1431
1432
1433
1434
1435
1436
1437
1438
1439
1440
1441
1442
1443
1444
1445
1446
1447
1448
1449
1450
1451
1452
1453
1454
1455
1456
1457

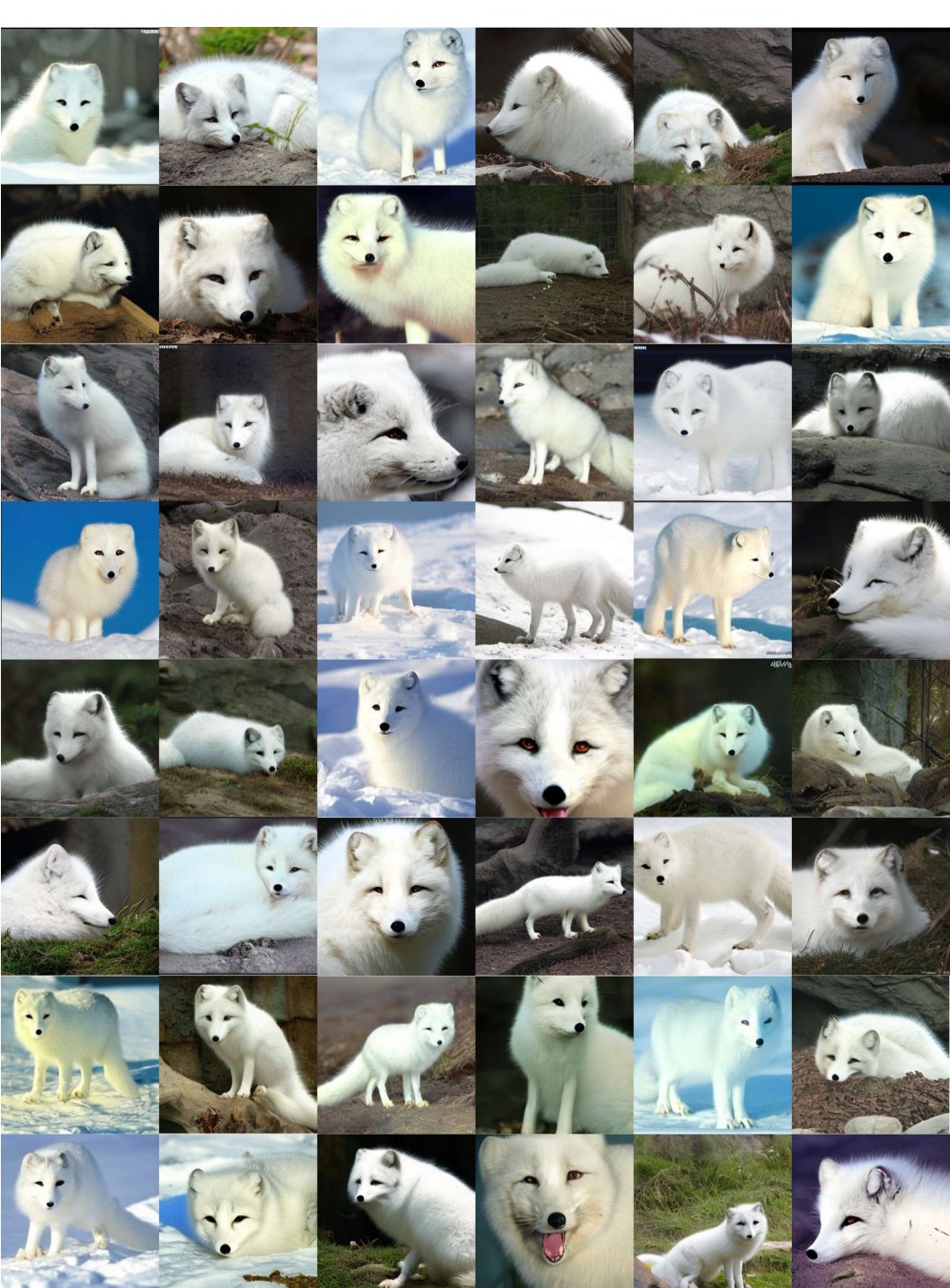

Figure 24: 256 × 256 samples of *ComboStoc*-XL/2 800K. Classifier-free guidance scale = 4.0. Class label = "Arctic fox, white fox, Alopex lagopus" (279).

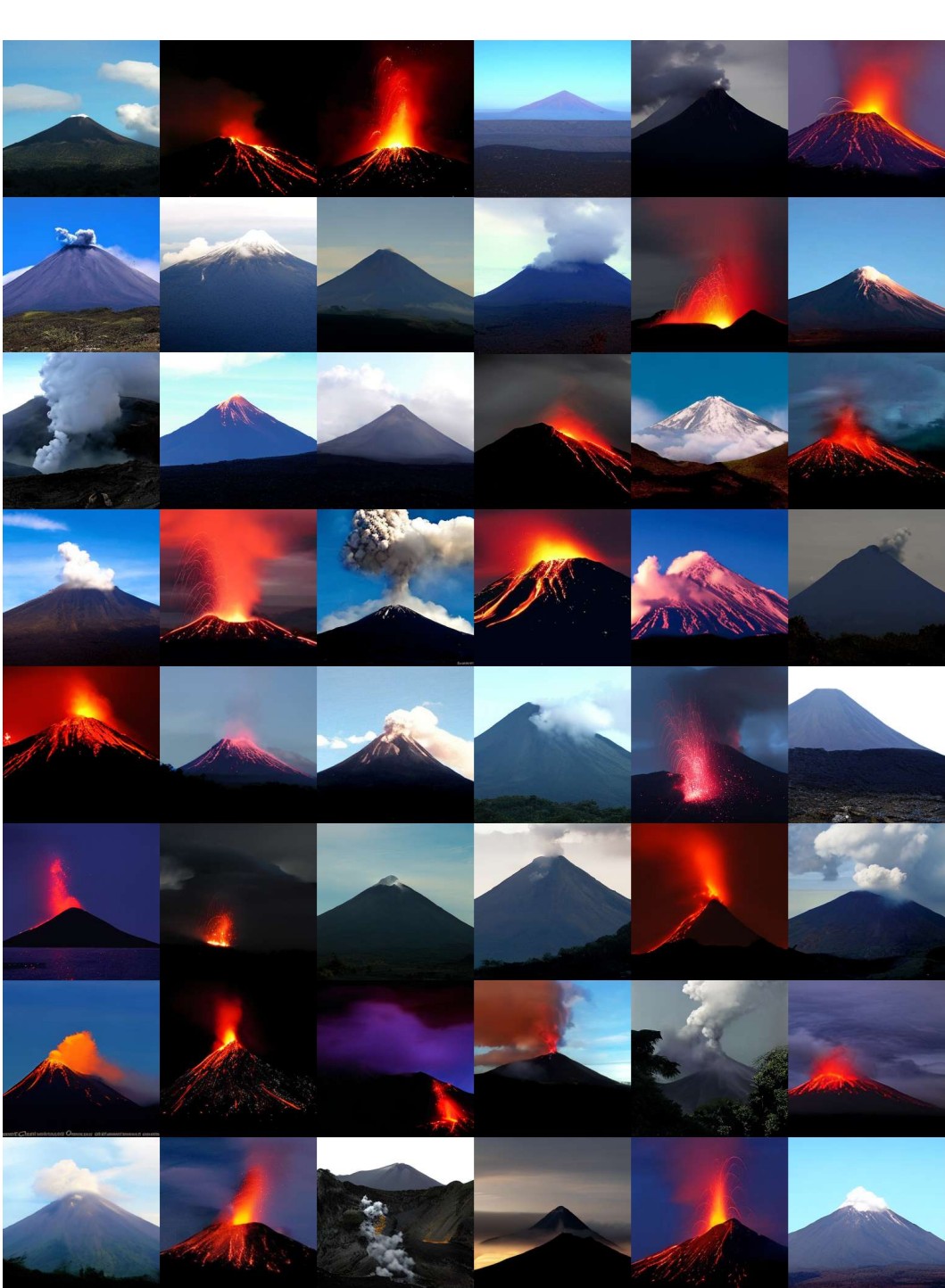

Figure 25: 256 × 256 samples of *ComboStoc*-XL/2 800K. Classifier-free guidance scale = 4.0. Class label = "volcano" (980).

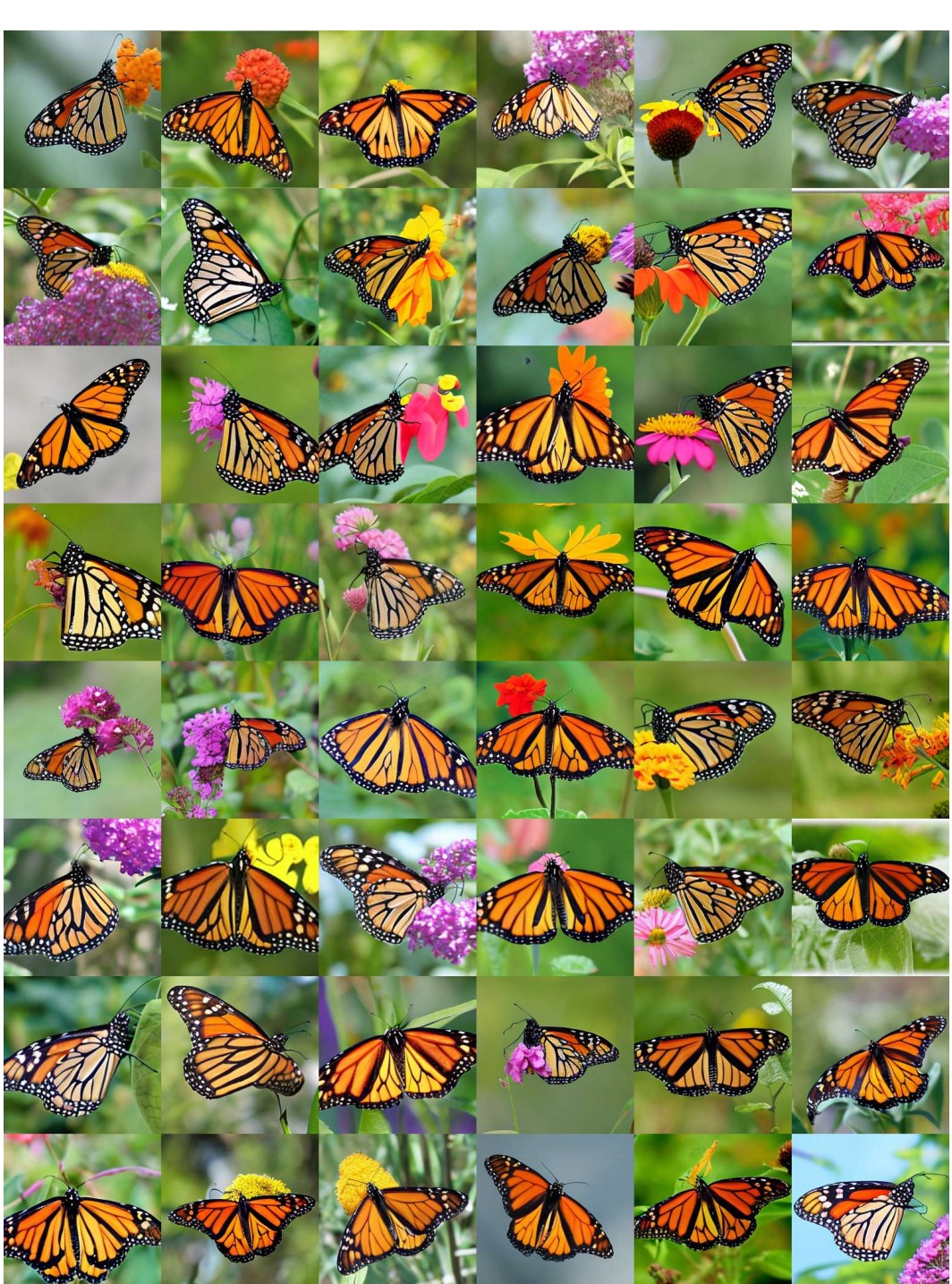

Figure 26: $256 \times 256$ samples of *ComboStoc*-XL/2 800K. Classifier-free guidance scale = 4.0. Class label = "monarch, monarch butterfly, milkweed butterfly, Danaus plexippus" (323).

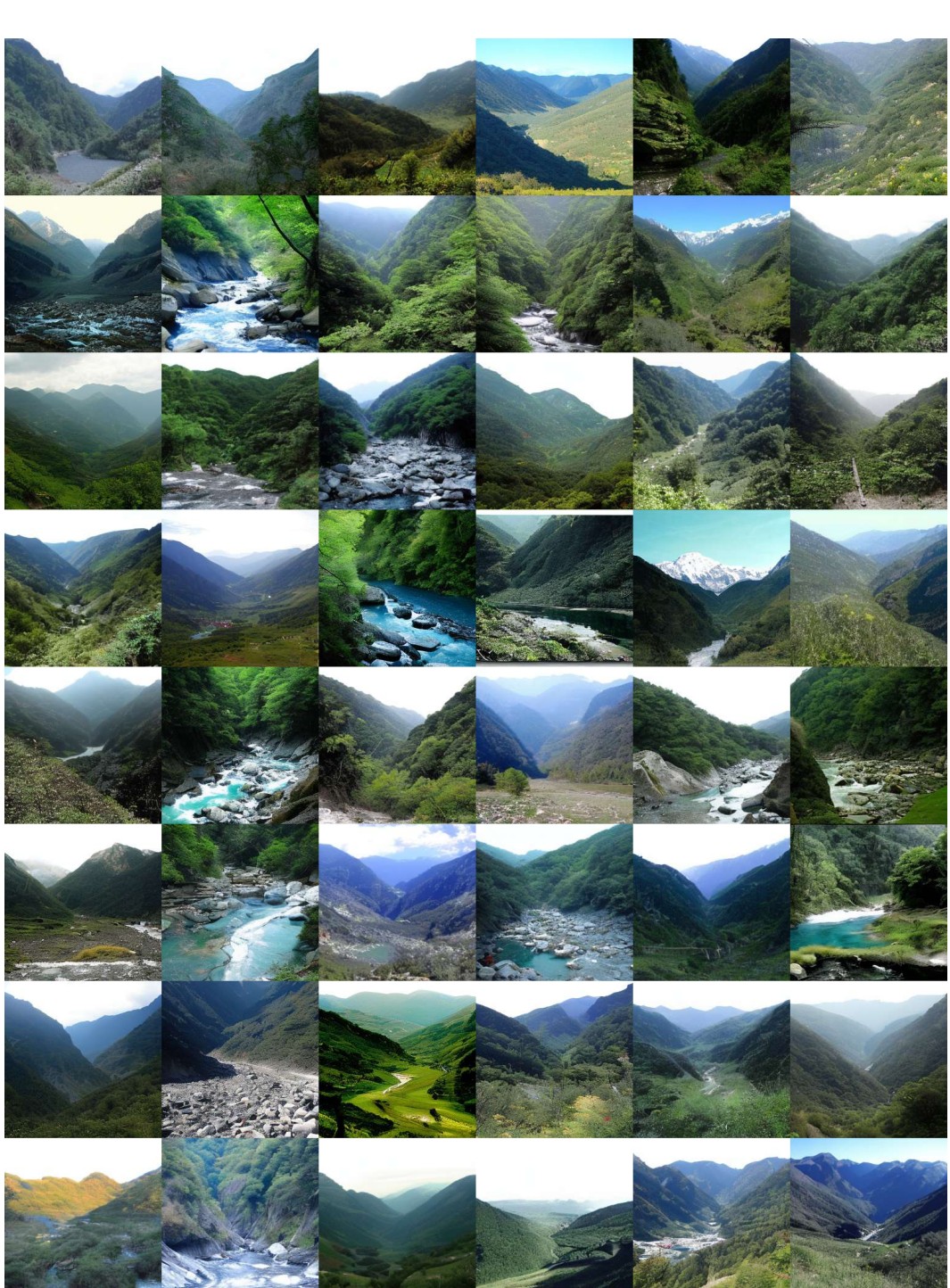

Figure 27: $256 \times 256$ samples of *ComboStoc*-XL/2 800K. Classifier-free guidance scale = 4.0. Class label = "valley, vale" (979).

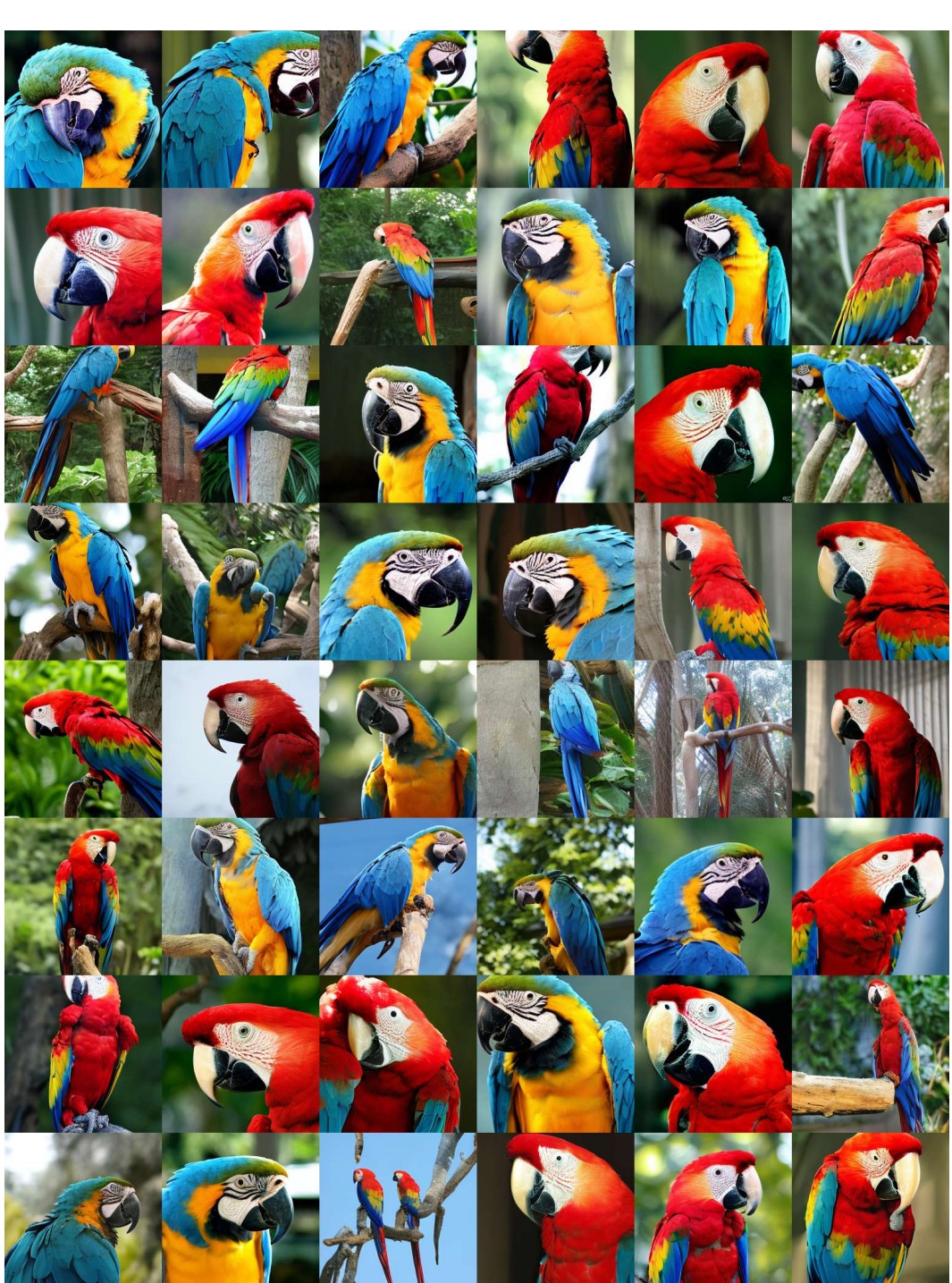

Figure 28: $256 \times 256$ samples of *ComboStoc*-XL/2 800K. Classifier-free guidance scale = 4.0. Class label = "macaw" (88).

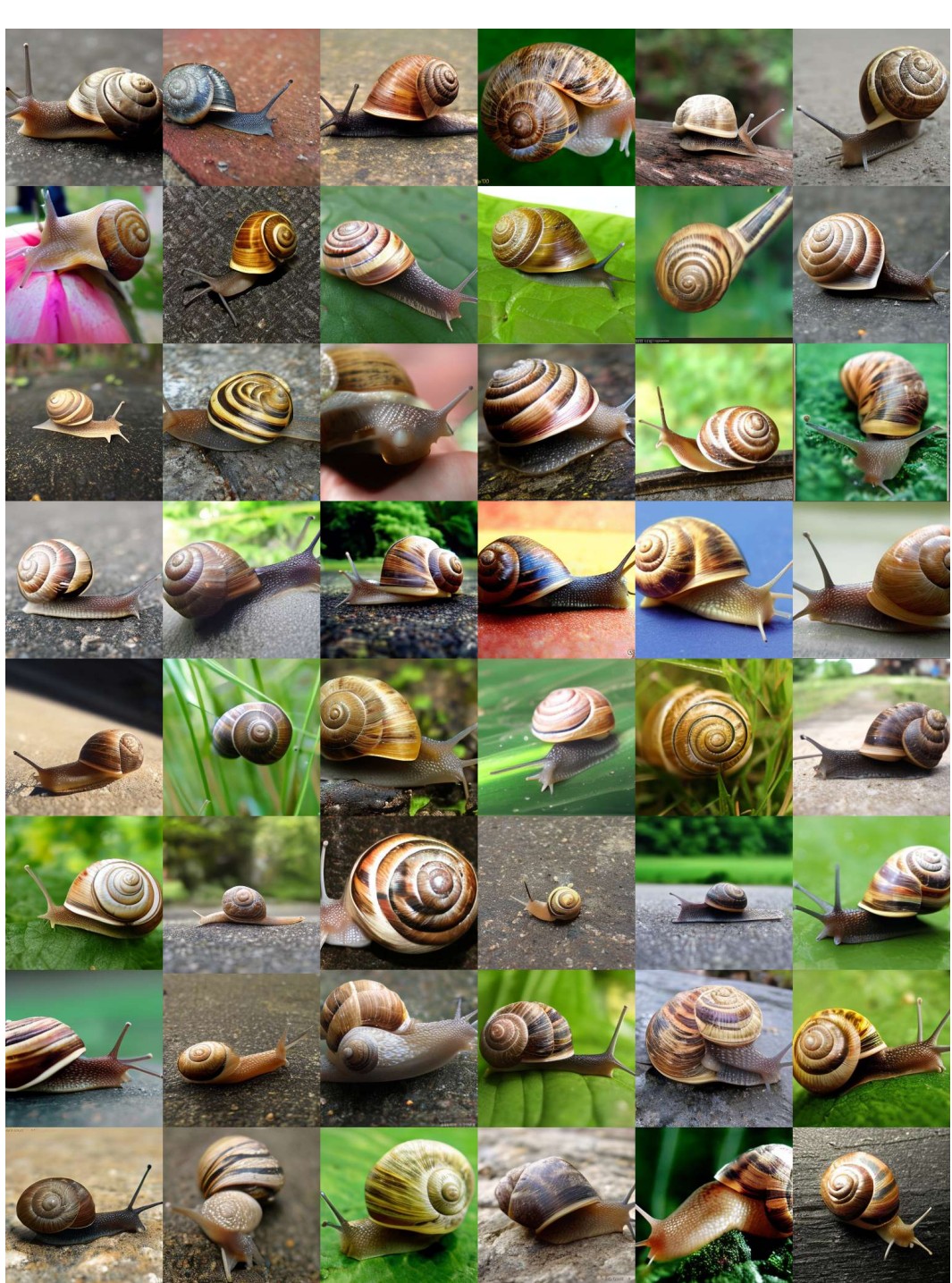

Figure 29: $256 \times 256$ samples of *ComboStoc*-XL/2 800K. Classifier-free guidance scale = 4.0. Class label = "snail" (113).

1728
1729
1730
1731
1732
1733
1734
1735
1736
1737
1738
1739
1740
1741
1742
1743
1744
1745
1746
1747
1748
1749
1750
1751
1752
1753
1754
1755
1756
1757
1758
1759
1760
1761
1762
1763
1764
1765
1766
1767
1768
1769
1770
1771
1772
1773
1774
1775
1776
1777
1778
1779
1780
1781

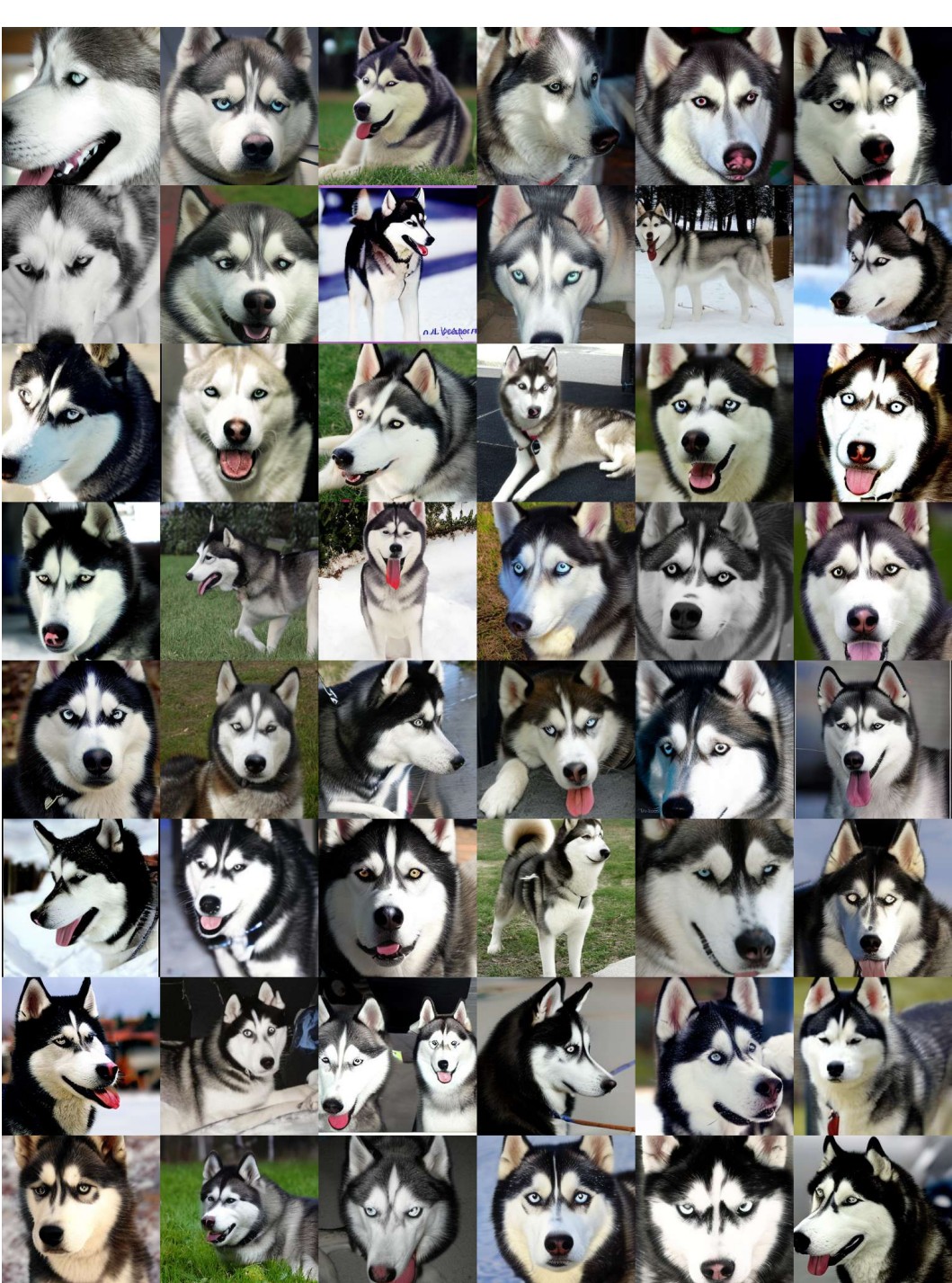

Figure 30: $256 \times 256$ samples of *ComboStoc*-XL/2 800K. Classifier-free guidance scale = 4.0. Class label = "Eskimo dog, husky" (248).

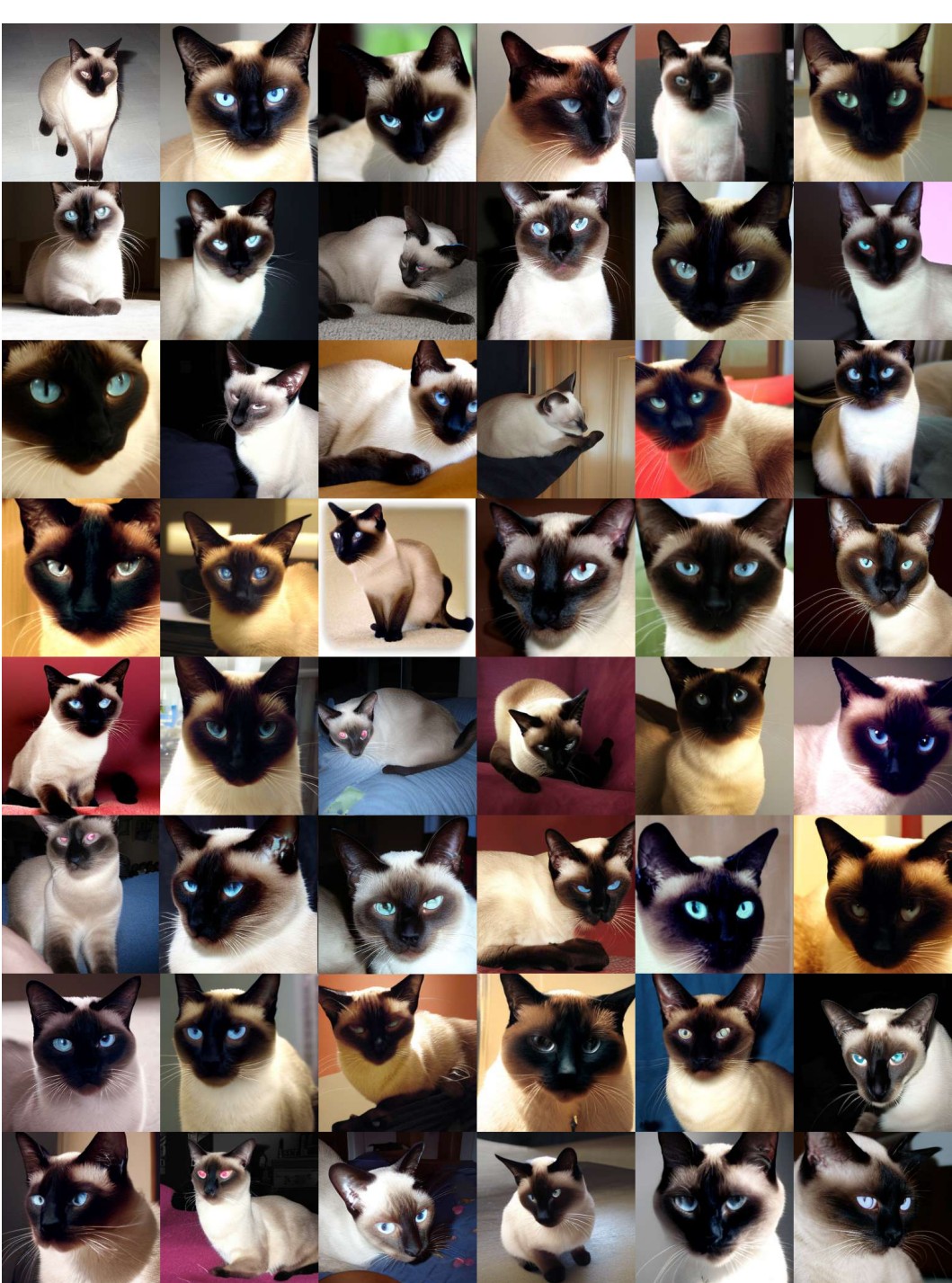

Figure 31: $256 \times 256$ samples of *ComboStoc*-XL/2 800K. Classifier-free guidance scale = 4.0. Class label = "Siamese cat, Siamese" (284).