# OpenReview forum: "ComboStoc: Combinatorial Stochasticity for Diffusion Generative Models"
_ICLR.cc/2025/Conference — ICLR 2025 Conference Withdrawn Submission_

### Official Review · Reviewer_H55J · 2024-10-18

**Soundness:** 4
**Presentation:** 4
**Contribution:** 4
**Rating:** 8
**Confidence:** 5

**Summary:**

This paper finds the combinatorial complexity problem of diffusion model training and proposes the ComboStoc method to train diffusion models, the core motivation of the author is that the existing interpolation methods lead to insufficient training of the model in the different timestep (i.e., $x_1|t=0.1. x_2|t=0.1$, in each training iter timestep is **same on all pixel**), so the authors design a new interpolation scheme in different pixel to improve it (i.e., $x_1|t=0.1. x_2|t=0.8$). Extensive experiments illustrate the effectiveness of the propsed method.

**Strengths:**

+ The combinatorial complexity in training of the diffusion model is not explored. This paper find this proboem and propose a sample way to simple fix to this problem. I think this find is really  interesting.
+ The author validat the method on both advanced DiT and SiT architectures, which fully illustrates the effectiveness of the methodology. Since SiT and DiT have different Noise Schedules, which also illustrates the robustness of the method.
+ The paper is well written and motivated with a clear description. And the code is given which makes the work very solid.

**Weaknesses:**

+ Some related works[1,2] have found contradictory gradients for different timesteps, while others[3, 4] find non-uniform variations of different timesteps during diffusion, which is of some relevance to this paper (improve the training of the diffusion models), and I suggest that the authors add this part of the discussion to the related work.

  [1] Addressing Negative Transfer in Diffusion Models NeurIPS-2023

  [2] Efficient Diffusion Training via Min-SNR Weighting Strategy ICCV-2023

  [3] A Closer Look at Time Steps is Worthy of Triple Speed-Up for Diffusion Model Training Arxiv-2024

  [4] Beta-Tuned Timestep Diffusion Model ECCV-2024

+ Figure 11 exemplifies the non-uniformity of the existing method, and the authors would be well advised to provide an optimized comparison plot, thus allowing the validity of the method to be better verified.

**Questions:**

The issues raised in this paper are very interesting, and I'm curious if the authors could do some experimentation or discussion of the more commonly used Noise Schedules (e.g., Coseine Schedules).

---

### Official Review · Reviewer_njbC · 2024-11-02

**Soundness:** 2
**Presentation:** 2
**Contribution:** 3
**Rating:** 5
**Confidence:** 3

**Summary:**

The paper shows that introducing multi-dimensional independent timesteps, each of which is responsible for different parts of input, e.g., per patch/feature basis, etc., can improve the overall training of diffusion models. Intuitively, this approach can help to extend the sampling coverage of input space during diffusion training, considering that the standard training only samples input along its corresponding diffusion trajectory. Experimental results on ImageNet (for image generation experiments) and PartNet (for 3D generation), mainly based on SiT, primarily report consistent improvements of FID. The paper also suggests a new application of the proposed method as a byproduct, viz., asynchronous reconstruction of input, which can be useful for image editing, fine-grained part control in 3D, etc.

**Strengths:**

- The paper is easy to follow.
- The proposed method is notably simple.
- The proposed method shows more effectiveness on structured 3D generation tasks (compared to well-studied ImageNet), an area where more study has been demanded.
- The experimental results are consistent in showing the effectiveness of the method.
- The application suggested from the method, i.e., asynchronous reconstruction, looks novel and has potential to be applied to other tasks.

**Weaknesses:**

- The overall experiments are limited to a single architecture, namely SiT. This raises a question whether the effectiveness of the method generalizes to other diffusion architectures.
- In case of ImageNet, the effectiveness of the method is solely supported by FID.
- Table 1: I think it is more fair to add SiT-XL (cfg=1.5) @ 800k result when evaluating ComboStoc (cfg=1.5) @ 800k.
- Discussions about computational efficiency are missing in the main text, while it seems important to have one as the method introduce extra complexity in diffusion model architectures (and may introduce overhead). For example, Figure 3 may also have a wall-clock comparison.
- Figure 8: There is no “baseline” results, thereby it is hard to judge whether the method indeed has effectiveness for the proposed application (or whether the standard diffusion model could just do the task as well). I think it would be also good to add some quantitative results in Section 4.2.
- It seems to me that at least some part of Appendix A.2 should be in the main text rather than Appendix. To my understanding, the “off-diagonal drift minimization” is a crucial component to enable the method to work, while its whole explanation is deferred to Appendix.

**Questions:**

- I am curious whether the effectiveness of the proposed method essentially depends on data scarcity, i.e., whether the gain is more significant in limited data regime. The under-sampling issue the paper described might become less important with enough data.
- Table 5: How could the proposed method maintain the same training/inference speed, despite of its increased GFLOPS, e.g., 237.34 → 352.46?

---

### Official Review · Reviewer_CPFT · 2024-11-02

**Soundness:** 3
**Presentation:** 2
**Contribution:** 3
**Rating:** 6
**Confidence:** 4

**Summary:**

The authors introduce ComboStoc, a new approach that allows diffusion models to better account for the combinatorial complexities in high-dimensional data. Traditional models often underperform because they do not explore all possible data combinations during training. In contrast, ComboStoc implements asynchronous time steps across different parts of the data during training to take advantage of nontrivial correlations. This method enables the model to fully explore the combinatorial space of features and attributes, thus, accelerating training and improving the quality of generated data in tasks involving images and 3D shapes compared to baseline models. Furthermore, the model gains better control over specific features during test time. This enables new generation methods, such as selectively modifying or controlling certain parts of an image or shape.

**Strengths:**

The idea of asynchronous time steps to enhance diffusion models is innovative and introduces a novel approach to accelerate the training of diffusion models.
The paper provides interesting experiments, especially in section 4.2 with the newly enabled applications, which demonstrate the effectiveness of their method for both image and 3D data.
The way they integrate the tensorized time step embedding with patch embedding is well-explained and integrated into the time embedding module of the SiT model.
The experiments are supported by clear visualizations, enhancing the understanding of the model's performance.
This approach appears to yield better qualitative and quantitative results compared to baseline models, without significantly increasing computational costs.

**Weaknesses:**

Some parts of the paper lack clarity and can be confusing to read.

For instance, it is somewhat unclear what the training algorithm looks like. How do you sample the asynchronous time steps at train time? How does it differ at test time? Do you have to apply the compensation drift at test time as well? Providing a clearer distinction between the training and test time setups would enhance understanding of how ComboStoc operates in typical use cases.
The paper could offer more intuition as to why asynchronous time steps are beneficial and why they encourage the network to learn the correlations of different dimensions and attributes.

There are some smaller grammatical errors (e.g., "each of the patch token is…").
Some abbreviations are introduced without explanation, which could confuse readers unfamiliar with them.
The sentence "To see why the space of possibilities has a combinatorial structure, we note that the data samples are most likely residing on high-dimensional spaces with clear combinatorial structures" is somewhat tautological.
The phrase "Correspondingly, the network fθ(xt) is trained to predict velocity, or the target data sample, etc." is vague, as it’s unclear when to predict velocity versus the target sample, and what "etc." refers to. However, it is good that you later specify that you predict velocity for image data and the target sample for 3D data.

**Questions:**

What does the training algorithm look like in practice, and what are the effects on training time?

Is tensorized time sampled at train time and/or at test time?

How relevant is asynchronous time sampling during test time?

By "during testing synchronized timesteps will be used if no graded control is applied", do you mean, that at test time, you generate new images completely from scratch and that all parts of the data share the same time step?

Is there a potential for spillover or contamination effects in these evaluations?

Is there an alternative approach to account for combinatorial complexity without relying on an asynchronous schedule? How does the asynchronous schedule specifically relate to managing combinatorial complexity?

Have you witnessed any sort of mode collapse and how diverse are the samples with ComboStoc?

Regarding the number of time steps in section 4.2, do all parts of the data receive the same number of steps, or is there variation? In specific, does the phrase “increase the time steps for individual dimensions and attributes via 1−t0 N for N steps” imply a rescaling of time steps between t0 and 1 so that these samples have the same number of denoising steps?

---

### Official Review · Reviewer_equh · 2024-11-03

**Soundness:** 3
**Presentation:** 3
**Contribution:** 2
**Rating:** 5
**Confidence:** 3

**Summary:**

This paper proposes ComboStoc, a simple but novel fix of diffusion generative models, which replaces the scalar timestep $t$ to tensor. This adjustment aims to effectively cover the entire path space, improving the model’s capacity to handle complex combinatorial structures. Experimental results on class-conditional image generation in ImageNet demonstrate clear improvement over baseline SiT models. The experiment on 3D part generation highlights that ComboStoc is well-suited for data with combinatorial structures. Moreover, ComboStoc also enables asynchronous denoising across different dimensions, allowing various generation scenarios such as image inpainting or 3d part assembling.

**Strengths:**

- The paper is well-written and easy to follow.
- The proposed modification is simple yet effective in exploring the entire path space, which has particular advantages for learning data with combinatorial structures.
- The clear improvements over strong baseline (SiT) models showcase the effectiveness and applicability across various generation tasks.

**Weaknesses:**

- The intuition behind the method is not clear. It is difficult to fully understand why insufficient coverage of low-density regions is critical in synchronous generation in previous approaches. When we follow a synchronous time schedule as in Figure 2(a), it will cover all the paths from $z \rightarrow x_1$. In this context, could the authors elaborate further on why these off-diagonal paths are problematic in synchronous generation?
- Although the proposed method demonstrates empirical effectiveness, it is still unclear why asynchronous timestep schedules should perform better than synchronous counterparts. L155-156 states that ”it encourages the network to learn the correlations of different dimensions and attributes…” but there is no further justification or empirical evidence on this claim. After reading this claim, it seems like such a scheduling strategy might have connections to masked modeling in generative models [1]. By varying noise levels across pixels (or sub-parts) in asynchronous timestep scheduling, the network may more effectively leverage correlations between sub-parts of the data to denoise the entire image. Moreover, this paper seems to provide a similar observation (Figure 3) to [1], where masked modeling accelerates learning structures and makes convergence much faster compared to standard diffusion training schemes. A clearer explanation of the claim in L155-156 would provide valuable insight and better understanding of the proposed method.
- While the proposed method is empirically effective, its technical novelty appears limited, as it essentially replaces a scalar timestep with a tensor representation.

[1] Gao et al., Masked Diffusion Transformer is a Strong Image Synthesizer, in ICCV 23.

**Questions:**

- The abstract mentions that the space is spanned by a combination of "dimensions" and "attributes," yet these terms lack clarity at first glance. It would improve readability if the authors could define these terms early in the paper, perhaps in the introduction, and include specific examples to help readers grasp the intended meaning of these concepts.
- Figure 7 indicates a lack of global consistency between the left and right sections of the images. For example, in the third row, when $\lambda > 0.5$, the colors differ on either side of the vertical center line and look unnatural. If ComboStoc effectively learns to denoise across different dimensions with asynchronous timesteps, shouldn’t the transition in the images be seamless?
- In L319, it states that half of the training batches in ImageNet are constructed synchronously, while this is not the case for 3D shape data. Could the authors elaborate more on the relationship between data size and the mixing scheme?
- In L181, the explanation about samples with asynchronous $\mathbf{t}$ should not predict the original velocity is unclear. Also, how do the two methods presented in Appendix A.2 differ? Are both methods regressing on $x_1 - x_{t_0}$​​ as the target velocity? Does this optimization ensure that the learned model captures the marginal velocity field in a manner similar to existing stochastic interpolation?

---

### Official Review · Reviewer_LAia · 2024-11-04

**Soundness:** 2
**Presentation:** 1
**Contribution:** 2
**Rating:** 3
**Confidence:** 3

**Summary:**

The paper proposes improving the diversity of the data distribution used to train stochastic interpolant models by leveraging the combinatorial nature of our data. The authors achieve this through sampling non-synchronised noised latents by taking the time component to be a vector. The authors then validate this approach experimentally on image and 3D shape generation tasks reporting improved performance.

**Strengths:**

The viewpoint of exploiting the combinatorial complexity of data for improving generative models is novel.

**Weaknesses:**

The paper is not well structured and the arguments given by the paper are also unclear.

1. The paper mainly considers the conditional interpolation paths for the motivation of their approach, however, the vector field learned by flow matching will in general not be linear as this is defined in terms on an expectation over all conditional vector fields. For example, in appendix a.2, the reasoning is in terms of integrating a linear vector field which doesn't happen in practice.
2. The terminology of the paper is unclear. The authors refer to the class of model they are working with as diffusion and stochastic interpolants in different places. They should be clear that they are using stochastic interpolants with the choice of the linear path and a Gaussian coupling.
3. The paper does not summarise their new training and sampling procedure, instead leaving the reader to infer this. For example, what exactly is the training objective used in the paper? For images, the authors training a neural network to predict the difference between the data and noise, which for 3D shapes, this predicts the data instead. Further, how is "off-diagonal drift" incorporated during training and sampling and what is the reasoning behind this approach?
3. The improvement reported by the authors could instead be due to a form of data augmentation enriching their data.
4. How is the form of \rho(x) derived in appendix A. This presentation seems to be non-standard.

**Questions:**

See weaknesses above.

---

### Note · Authors · 2024-11-15

**Comment:**

We thank all the reviewers for their helpful suggestions and questions. Given the predominantly negative feedback from the reviewers, we have decided to withdraw this paper.

**Withdrawal Confirmation:**

I have read and agree with the venue's withdrawal policy on behalf of myself and my co-authors.